# FORMAL MECHANISTIC INTERPRETABILITY: AUTOMATED CIRCUIT DISCOVERY WITH PROVABLE GUARANTEES

**Itamar Hadad**[1]**, Guy Katz**[1]**, Shahaf Bassan**[1]

School of Computer Science and Engineering, The Hebrew University of Jerusalem[1]

`itamar.hadad@mail.huji.ac.il, g.katz@mail.huji.ac.il,`
`shahaf.bassan@mail.huji.ac.il`

## ABSTRACT

*Automated circuit discovery* is a central tool in mechanistic interpretability for identifying the internal components of neural networks responsible for specific behaviors. While prior methods have made significant progress, they typically depend on heuristics or approximations and do not offer provable guarantees over continuous input domains for the resulting circuits. In this work, we leverage recent advances in neural network verification to propose a suite of automated algorithms that yield circuits with *provable guarantees*. We focus on three types of guarantees: (i) *input domain robustness*, ensuring the circuit agrees with the model across a continuous input region; (ii) *robust patching*, certifying circuit alignment under continuous patching perturbations; and (3) *minimality*, formalizing and capturing a wide array of various notions of succinctness. Interestingly, we uncover a diverse set of novel theoretical connections among these three families of guarantees, with critical implications for the convergence of our algorithms. Finally, we conduct experiments with state-of-the-art verifiers on various vision models, showing that our algorithms yield circuits with substantially stronger robustness guarantees than standard circuit discovery methods — establishing a principled foundation for provable circuit discovery.

## 1 INTRODUCTION

The rapid rise of neural networks, driven by transformative architectures such as Transformers, has reshaped both theory and applications. Alongside this revolution, *interpretability* has become a central research direction (Zhang et al., 2021; Räuker et al., 2023); and more recently, efforts have focused on *mechanistic interpretability* (MI), which aims to reverse-engineer neural networks into human-understandable components and functional modules (Olah et al., 2020; Olah, 2022; Zhao et al., 2024). MI offers fine-grained interpretability that serves various purposes, including transparency, trustworthiness, safety, and other applications (Bereska & Gavves, 2024; Zhou et al., 2025).

A central open challenge in MI is *circuit discovery* (Olah et al., 2020), which seeks to identify subgraphs within neural networks, called *circuits*, that drive specific model behaviors. Recent works propose varied approaches (Wang et al., 2023; Conmy et al., 2023; Rajaram et al., 2024), differing by domain (text vs. vision), patching methods (zero, mean, sampling), and the balance between manual and automated steps. However, despite substantial progress, most current circuit discovery algorithms remain heuristic or approximate, without rigorous guarantees of circuit faithfulness, particularly under *continuous* perturbation domains (Adolfi et al., 2025; Miller et al., 2024; Méloux et al., 2025). This limitation is concerning: even small perturbations can break circuit faithfulness, and since circuit discovery is tied to *safety* considerations (Bereska & Gavves, 2024), such guarantees are essential.

**Our Contributions.** To address these concerns, we introduce a novel algorithmic framework that builds on recent and exciting advances in the emerging field of *neural network verification* (Wang et al., 2021; Zhou et al., 2024; Brix et al., 2024; Kotha et al., 2023; Ferrari et al., 2022), enabling the derivation of circuits with provable guarantees across continuous domains of interest.

## 1.1 THEORETICAL CONTRIBUTIONS

- We formalize a set of novel provable guarantees for circuit discovery that hold strictly over *entire continuous domains*. These include: (i) *input-domain robustness*, ensuring circuits remain faithful across continuous input regions; (ii) *patching-domain robustness*, addressing criticisms of sampling-based ablation; and (iii) a broad family of *minimality guarantees*, extending earlier notions to include *quasi-*, *local-*, *subset-*, and *cardinal-*minimality.

- We present novel theoretical proofs that reveal strong connections between these three families of guarantees. At the core is the *circuit monotonicity* property, which underpins minimality guarantees for optimization algorithms and clarifies the conditions under which they hold. We also establish a crucial *duality* between circuits and small "blocking" subgraphs, enabling the efficient discovery of circuits with much stronger minimality guarantees.

## 1.2 EMPIRICAL CONTRIBUTIONS

- We propose a framework for encoding both input- and patching-robustness guarantees in neural networks and their circuits, using a technical *siamese encoding* of the network with its associated circuit or patching-domain, which enables certifying the desired properties.

- We introduce a set of novel automated algorithms that preserve the invariants of the robustness guarantees and prove that each converges to circuits meeting our various minimality criteria. These algorithms enable a trade-off between computational cost and the degree of minimality achieved in the resulting circuits.

- We conduct extensive experiments with $\alpha$–$\beta$-CROWN, the state-of-the-art in neural network verification, to derive circuits with input, patching, and minimality guarantees. These are evaluated on standard neural vision models commonly used in the neural network verification literature. Compared to sampling-based approaches, our framework certifies robustness, whereas even infinitesimal perturbations break the faithfulness of sampling-based circuits.

Overall, we believe these contributions mark a significant step forward in establishing both theoretical and empirical foundations for circuit discovery with provable guarantees, paving the way for a wide range of future research directions.

## 2 PRELIMINARIES

### 2.1 NOTATION

Let $f_G : \mathbb{R}^n \to \mathbb{R}^d$ denote a neural network, with $G := \langle V, E \rangle$ representing its computation graph. The precise structure of $G$ — that is, what each node and edge correspond to (e.g., neurons, attention heads, positional embeddings, convolution filters) — is determined both by the network's architecture and by the level of granularity chosen by the user. A *circuit* $C$ is defined as a subgraph $C \subseteq G$, consisting of components hypothesized to drive the model's behavior on a task. Each such circuit naturally induces a function $f_C : \mathbb{R}^n \to \mathbb{R}^d$, obtained by restricting $f_G$ to the components in $C$.

In circuit discovery, the complement $\overline{C} := G \setminus C$ is often fixed to constant activations, a practice known as *patching*, with variants such as zero-patching or mean-patching. Let $f_G$ have $L \in \mathbb{N}$ layers with activation spaces $V_i$ for $i \in [L]$, and let $\mathcal{X} \subseteq \mathbb{R}^n$ be an input domain. For $\mathbf{x} \in \mathcal{X}$, denote the activations at layer $i$ as $h_i(\mathbf{x}) \in V_i$, and the reachable activation space as $\mathcal{H}_G(\mathcal{X}) = \{(h_1(\mathbf{x}), \ldots, h_L(\mathbf{x})) : \mathbf{x} \in \mathcal{X}\} \subseteq V_1 \times \cdots \times V_L$. We write $\mathcal{H}_{\overline{C}}(\mathcal{X})$ for the *partial* reachable activation space over $\overline{C}$. For a partial activation assignment $\alpha \in \mathcal{H}_{\overline{C}}(\mathcal{X})$, we write $f_C(\mathbf{x} \mid \overline{C} = \alpha)$ to denote inference through $f_C(\mathbf{x})$, constructed from the components of $C$, while fixing the activations of $\overline{C}$ to the values in $\alpha$.

### 2.2 NEURAL NETWORK VERIFICATION

Consider a generic neural network $f_G$ with arbitrary element-wise nonlinear activations. Many tools exist to formally verify properties of such networks, with adversarial robustness being the most studied (Brix et al., 2024). Formally, the neural network verification problem can be stated as follows:

---

**Neural Network Verification (Problem Statement)**:
**Input**: A neural network model $f_G$, for which $\mathbf{y} := f_G(\mathbf{x})$, with an input specification $\psi_{\text{in}}(\mathbf{x})$, and an *unsafe* output specification $\psi_{\text{out}}(\mathbf{y})$.
**Output**: *No*, if there exists some $\mathbf{x} \in \mathbb{R}^n$ such that $\psi_{\text{in}}(\mathbf{x})$ and $\psi_{\text{out}}(\mathbf{y})$ both hold, and *Yes* otherwise.

---

A variety of off-the-shelf neural network verifiers have been developed (Brix et al., 2024). When the input constraints $\psi_{in}(\mathbf{x})$, output constraints $\psi_{out}(\mathbf{y})$, and model $f_G$ are piecewise-linear (e.g., ReLU activations), the verification problem can be solved exactly (Katz et al., 2017). In practice, it is often relaxed for efficiency, and the output is enclosed within bounds that account for approximation errors.

## 3 PROVABLE GUARANTEES FOR CIRCUIT DISCOVERY

### 3.1 INPUT DOMAIN GUARANTEES

In this subsection, we introduce the first set of guarantees that our algorithms are designed to satisfy — specifically, *provable robustness* against input perturbations. A central challenge when evaluating a circuit's faithfulness is that, even if it matches the model at one point or a finite set of points, small input perturbations can quickly break this agreement. To overcome this limitation, our first definition considers circuits that are not only faithful to the model on a discrete set of points but are also *provably robust* across an entire infinite *continuous set* of inputs.

**Definition 1** (Input-robust circuit). *Given a neural network $f_G$ and a union of continuous domains $\mathcal{Z} \subseteq \mathbb{R}^n$ (e.g., a union of $\ell_p$-balls of radius $\epsilon_p$ around a set of discrete points $\{\boldsymbol{x}_j\}_{j=1}^k$), we say that a subgraph $C \subseteq G$ is an* input-robust circuit *with respect to $\langle f_G, \mathcal{Z} \rangle$, a fixed patching vector $\alpha$ applied to the complement set $\overline{C} := G \setminus C$, and a tolerance level $\delta \in \mathbb{R}^+$, if and only if:*

$$\forall \boldsymbol{z} \in \mathcal{Z} := \bigcup_{j=1}^k \mathcal{B}_{\epsilon_p}^p(\boldsymbol{x}_j). \quad \|f_C(\boldsymbol{z} \,|\, \overline{C} = \alpha) - f_G(\boldsymbol{z}))\|_p \leq \delta,$$

$$s.t. \; \mathcal{B}_{\epsilon_p}^p(\boldsymbol{x}_j) := \{\boldsymbol{z}_j \in \mathbb{R}^n \mid \|\boldsymbol{z}_j - \boldsymbol{x}_j\|_p \leq \epsilon_p\} \tag{1}$$

**Certifying circuit input robustness via verification.** Neural network verification properties are typically encoded over a *single* model $f_G$, while the circuit input robustness property (Def. 1) requires evaluating both the model graph $G$ and a circuit $C \subseteq G$. To address this, we introduce a novel method to certify the property in Def. 1 using a *siamese encoding* of the network $f_G$. Specifically, we duplicate the circuit $C' := C$ and "stack" it with $G$ to form a combined model $G \sqcup C'$ with a *shared* input layer. This induces a function $f_{(G \sqcup C')} : \mathbb{R}^n \to \mathbb{R}^{2d}$. The activations of the non-circuit components in the duplicate, $\overline{C'} := G \setminus C'$, are fixed to a constant $\alpha$, so for any $\mathbf{z} \in \mathbb{R}^n$

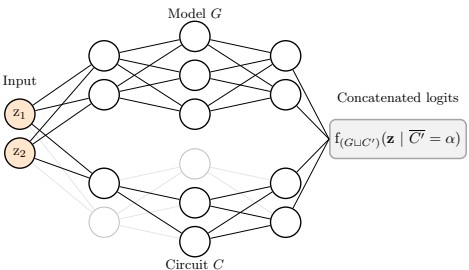

Figure 1: Illustration of the Siamese encoding for certifying the guarantee in Def. 1.

inference is $f_{(G \sqcup C')}(\mathbf{z} \,|\, \overline{C'} = \alpha)$, enabling direct certification of Def. 1 over the *combined* model. The input constraint $\psi_{in}(\mathbf{x})$ bounds $\mathbf{x}$ within $\mathcal{Z}$, while the output constraint $\psi_{out}(\mathbf{y})$ bounds the distance measure between the logits of $C'$ and $G$. Further details of this encoding appear in Appendix E.

### 3.2 PATCHING DOMAIN GUARANTEES

A central challenge in circuit discovery lies in deciding how to assign values to the complementary activations of a circuit — a process known as *patching*. The goal of patching is to replace these values to isolate the circuit's contribution. Prior work has examined several approaches, including zero-patching, which has been criticized as arbitrary since such values may be out-of-distribution if unseen during training (Conmy et al., 2023; Wang et al., 2023). Other strategies include mean-value patching (Wang et al., 2023) and sampling from discrete input distributions (Conmy et al., 2023). Yet, these methods still rely on evaluating complementary activations over a *discrete* set of sampled inputs, which may fail to generalize in *continuous* domains: even small perturbations in the

patching scheme can undermine a circuit's faithfulness. By analogy to *input*-robustness, we introduce *patching*-robustness: the requirement that a circuit preserve its faithfulness across an entire provable range of feasible perturbations to the complementary activations over a continuous input domain.

**Definition 2** (Patching-robust circuit). *Given a neural network $f_G$, a continuous input domain $\mathcal{Z} \subseteq \mathbb{R}^n$, and a reference set of inputs $\{\boldsymbol{x}_j\}_{j=1}^k \subseteq \mathcal{X}$, we say that $C \subseteq G$ is a* patching-robust circuit *with respect to $\langle f_G, \mathcal{Z} \rangle$ and a tolerance level $\delta \in \mathbb{R}^+$, iff for every $\boldsymbol{x}_j$ in $\{\boldsymbol{x}_j\}_{j=1}^k$:*

$$\forall \alpha \in \mathcal{H}_{\overline{C}}(\mathcal{Z}). \quad \|f_C(\boldsymbol{x}_j \mid \overline{C} = \alpha) - f_G(\boldsymbol{x}_j)\|_p \leq \delta \tag{2}$$

**Certifying circuit patching robustness via verification.** Analogous to input robustness, we introduce here a novel method to certify the property in Def 2, using a siamese encoding of $f_G$. Concretely, we duplicate the circuit $C' := C$ and "stack" it with $G$, but now $G$ and $C'$ have disjoint input domains, yielding $f_{(G \sqcup C')} : \mathbb{R}^{2n} \to \mathbb{R}^d$. We connect $C'$ and $G$ by fixing the *activations* of $\overline{C'}$ to those attained by $\mathcal{H}_{\overline{C}}(\mathbf{z})$ when evaluating $f_G(\mathbf{z})$. Thus, inference for $(\mathbf{x}, \mathbf{z}) \in \mathbb{R}^{2n}$ is given by $f_{(G \sqcup C')}(\mathbf{x} \mid \overline{C'} = \mathcal{H}_{\overline{C}}(\mathbf{z}))$. We then set input constraints to bound $\mathbf{z}$ within $\mathcal{Z}$, and output constraints to limit the distance $\|\cdot\|_p$ between the logits of $C'$ and $G$. Further details appear in Appendix F. We remark that input robustness and patching robustness can also be certified simultaneously within a single verification query by extending the siamese encoding (see Appendix G).

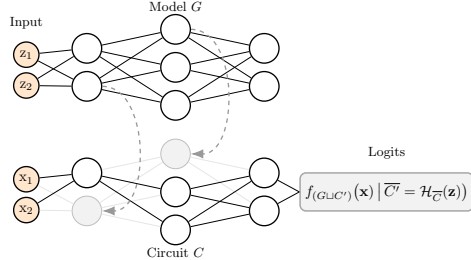

Figure 2: Illustration of the Siamese encoding for certifying the guarantee in Def. 2. Gray neurons denote non-circuit units whose activations are patched with those of the full model.

## 4 FROM CIRCUITS TO MINIMAL CIRCUITS

A common convention in the literature is that *smaller* circuits (i.e., lower circuit size) are generally considered more interpretable than larger ones (Mueller et al., 2025; Adolfi et al., 2025; Chowdhary et al., 2025; Wang et al., 2023; Bhaskar et al., 2024; Shi et al., 2024; Conmy et al., 2023). This makes *minimality* an important additional guarantee (Adolfi et al., 2025; Chowdhary et al., 2025; Shi et al., 2024; Mueller et al., 2025). While many works have pursued minimal circuits, recent studies highlight that "minimality" itself can take different forms (Adolfi et al., 2025), ranging from the weak notion of *quasi*-minimality to the strong notion of *cardinal*-minimality. In this work, we extend this spectrum to four main forms and provide rigorous proofs linking them to different optimization algorithms.

### 4.1 MINIMALITY GUARANTEES

In this subsection, we introduce four central notions of minimality. Since minimality must be defined relative to what qualifies as a "valid" or faithful circuit, we begin by specifying a *general faithfulness predicate*, $\Phi(C, G)$. Given a circuit $C \subseteq G$ within the computation graph $G$ of a model $f_G$, $\Phi(C, G)$ returns *True* if $C$ is faithful under some condition of interest, and *False* otherwise. Instances of $\Phi$ may include standard sampling-based measures used in circuit discovery, such as requiring the mean-squared error or KL-divergence between $C$ and $G$ to remain below a threshold $\tau$ (Conmy et al., 2023). Alternatively, $\Phi$ can reflect our *provable* measures that hold across continuous domains (Section 3), e.g., defining $\Phi(C, G)$ to require input and/or patching robustness.

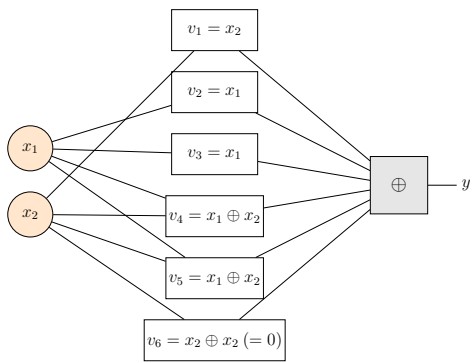

Figure 3: A toy Boolean circuit.

Consider, for example, a toy Boolean circuit presented in Fig. 3 as a running example. For simplicity, we assume that each node in the circuit corresponds to a component of the model.

The network takes inputs $(x_1, x_2) \in \{0,1\}^2$, whose outputs are aggregated by XOR, yielding $f_G(x_1, x_2) = v_1 \oplus v_2 \oplus \cdots \oplus v_6 = x_2$, since

$$f_G(x_1, x_2) := x_2 \oplus x_1 \oplus x_1 \oplus (x_1 \oplus x_2) \oplus (x_1 \oplus x_2) \oplus (x_2 \oplus x_2)$$
$$= x_2 \oplus \underbrace{(x_1 \oplus x_1)}_{=0} \oplus \underbrace{[(x_1 \oplus x_2) \oplus (x_1 \oplus x_2)]}_{=0} \oplus \underbrace{(x_2 \oplus x_2)}_{=0} = x_2.$$

For simplicity, we set the faithfulness predicate to consider the very strong condition of strict equality between the circuit and the model for *every* boolean input. In other words, for all $x_1$ and $x_2$ in $\{0,1\}^2$, it holds that $\Phi(C, G) := f_C(x_1, x_2) = f_G(x_1, x_2) = x_2$. A more detailed description of this example appears in Appendix C.3.

We begin with the weakest form of minimality and proceed step by step until we reach the strongest. The first notion, *quasi*-minimality, introduced by (Adolfi et al., 2025), defines $C$ as a subset that includes a "breaking point": *some* component that, when removed, breaks the circuit's faithfulness:

**Definition 3** (Quasi-Minimal Circuit). *Given $G$, a* quasi-minimal *circuit $C \subseteq G$ concerning $\Phi$, is a circuit for which $\Phi(C, G)$ holds ("$C$ is faithful"), and there exists some element (i.e., node/edge) $i$ in $C$ for which $\Phi(C \setminus \{i\}, G)$ is false.*

In our running example $C = \{v_1, \ldots, v_5\}$ is quasi-minimal: it satisfies $v_1 \oplus \cdots \oplus v_5 = f_G$, but removing *some* essential node (e.g., $v_1$) breaks this equality. However, we observe that the *quasi*-minimality notion of (Adolfi et al., 2025) can be strengthened: instead of requiring a single breaking point, one can demand that *every* component serves as one. Following conventions in the optimization literature, we refer to this stronger notion as *local* minimality.

**Definition 4** (Locally-Minimal Circuit). *Given $G$, a* locally-minimal *circuit $C \subseteq G$ concerning $\Phi$, is a circuit for which $\Phi(C, G)$ holds, and for any element $i$ in $C$, $\Phi(C \setminus \{i\}, G)$ is false.*

In our example, $C = \{v_1, v_2, v_3\}$ is locally minimal, as the predicate holds for $\{v_1, v_2, v_3\}$, but does not hold for $\{v_1, v_2\}$, $\{v_1, v_2\}$, and $\{v_2, v_3\}$. However, while local-minimality is stronger than quasi-minimality, it still has a limitation. Although removing any *single* component from the circuit $C$ breaks it, removing *multiple* components may still leave $C$ valid, giving a misleading sense of minimality. For instance, in our locally minimal circuit $C = \{v_1, v_2, v_3\}$, observe that while removing any single component (i.e., $v_1$, $v_2$, or $v_3$) breaks faithfulness, removing the *pair* $\{v_2, v_3\}$ (i.e., keeping only $v_1$) nonetheless results in a faithful circuit. To address this, we define a stronger notion: *subset-minimality*, which requires every *subset* of components to be a breaking point.

**Definition 5** (Subset-Minimal Circuit). *Given $G$, a* subset-minimal *circuit $C \subseteq G$ concerning $\Phi$, is a circuit for which $\Phi(C, G)$ holds true, and for any subgraph $S \subsetneq C$, $\Phi(C \setminus S, G)$ is false.*

In the running example, $C = \{v_2, v_4\}$ is a subset-minimal circuit, as $v_2 \oplus v_4 = x_1 \oplus (x_1 \oplus x_2) = x_2$, since every strict subset ($\{v_2\}, \{v_4\}$) fails to compute $f_G$. We note that even this significantly stronger notion of subset-minimality does not necessarily yield subsets of the absolute lowest cardinality. To address this limitation, the final notion introduces the strongest form: a *cardinally-minimal* circuit, which corresponds to the global optimum of minimality.

**Definition 6** (Cardinally-Minimal Circuit). *Given $G$, a* cardinally-minimal *circuit $C \subseteq G$ concerning $\Phi$ is a circuit for which $\Phi(C, G)$ is true, and has the lowest cardinality $|C|$ (i.e., there is no circuit $C' \subseteq G$ for which $\Phi(C', G)$ is true and $|C'| < |C|$).*

Here, $\{v_1\}$ is cardinally minimal, since $v_1 = x_2$ is functionally equivalent to $f_G$ and no smaller faithful circuit exists.

## 4.2 Algorithms for local and quasi minimal circuits

In this subsection, we present optimization algorithms for discovering circuits with provable guarantees. Building on prior circuit discovery frameworks, we show how modifying or validating optimization objectives changes the resulting guarantees. We also establish theoretical links between objectives based on continuous robustness guarantees and different notions of minimality. We begin with a standard greedy algorithm (Algorithm 1):

---

**Algorithm 1** Greedy Circuit Discovery Iterative Search

---

1: **Input** Model $f_G$, circuit faithfulness predicate $\Phi$
2: $C \leftarrow G$ under some given element ordering (e.g., reverse topological sort)
3: **for all** $i \in C$ **do**
4:     **if** $\Phi(C \setminus \{i\}, G)$ **then**
5:        $C \leftarrow C \setminus \{i\}$
6:     **end if**
7: **end for**
8: **return** $C$

---

Algorithm 1 describes a standard greedy procedure that starts from the full model graph $G$ and iteratively removes components as long as the faithfulness predicate holds. While this structure underlies many circuit discovery methods, we also consider an exhaustive variant, given as Algorithm 3 in Appendix C, which repeatedly iterates over the remaining components and attempts further removals, stopping only when every *single* component is critical. This guarantees the following:

**Proposition 1.** *Given any model $f_G$ and faithfulness predicate $\Phi$, Algorithm 1 visits a* quasi-minimal *circuit and Algorithm 3 converges to a* locally-minimal *circuit $C$, both with respect to $\Phi$.*

We note that each evaluation of $\Phi(C \setminus \{i\}, G)$ may be costly, depending on the predicate $\Phi$ (e.g., certifying input or patching robustness). To mitigate this, one may use a lighter notion of minimality — the *quasi*-minimal circuits of (Adolfi et al., 2025) — which require only a logarithmic, rather than linear, number of invocations. For completeness, we formally present and extend their binary-search procedure, provided as Algorithm 4 in Appendix C. Algorithm 4 follows a procedure similar to Algorithm 1, but employs a binary rather than iterative search. As a result, it yields a weaker notion of minimal circuits, while requiring fewer queries. We can therefore establish the following proposition:

**Proposition 2.** *While Algorithm 4 converges to a quasi-minimal circuit using $\mathcal{O}(\log |G|)$ evaluations of $\Phi(C, G)$ (Adolfi et al., 2025), Algorithm 1 visits a quasi-minimal circuit with $\mathcal{O}(|G|)$ evaluations, and Algorithm 3 converges to a locally-minimal circuit using $\mathcal{O}(|G|^2)$ evaluations.*

Finally, we note that Algorithms 1, 3 and 4 converge only to relatively "weak" forms of minimality. Even the stronger local-minimality guarantee of Algorithm 3 can fall short: while every single-element removal $C \setminus \{i\}$ breaks faithfulness, removing two elements simultaneously, $C \setminus \{i, j\}$, may still yield a faithful circuit. This undermines $C$'s "minimality" and shows that neither algorithm ensures the stronger notion of *subset*-minimality.

**Proposition 3.** *There exist infinitely many configurations of $f_G$, and $\Phi$, for which Algorithms 1, 3 and 4* do not *converge to a subset-minimal circuit $C$ concerning $\Phi$.*

### 4.3 THE CIRCUIT MONOTONICITY PROPERTY AND ITS IMPACT ON MINIMALITY

To address the issue of algorithms converging to "bad" local minima, we identify a key property of the faithfulness predicate $\Phi$ with crucial implications for stronger minimal subsets — *monotonicity*:

**Definition 7.** *We say that a circuit faithfulness predicate $\Phi$ is* monotonic *iff for any $C \subseteq C' \subseteq G$ it holds that if $\Phi(C, G)$ is true, then $\Phi(C', G)$ is true.*

Intuitively, monotonicity means that once $\Phi(C, G)$ holds for a circuit $C$ (i.e., it is "faithful"), it will keep holding as elements are added. In other words, enlarging the circuit never breaks faithfulness. This property is essential for Algorithm 1, as it underpins the stronger minimality guarantee:

**Proposition 4.** *If $\Phi$ is monotonic, then for any model $f_G$, Algorithm 1 converges to a subset-minimal circuit $C$ concerning $\Phi$.*

**The condition of monotonicity.** Interestingly, we establish a novel connection between the guarantees on the input and patching domains outlined in Section 3 and the monotonic behavior of $\Phi$:

**Proposition 5.** *Let $\Phi(C, G)$ denote validating whether $C$ is input-robust concerning $\langle f_G, \mathcal{Z} \rangle$ (Def. 1), and simultaneously patching-robust concerning $\langle f_G, \mathcal{Z}' \rangle$ (Def. 2). Then if $\mathcal{Z} \subseteq \mathcal{Z}'$ and $\mathcal{H}_G(\mathcal{Z}')$ is closed under concatenation, $\Phi$ is monotonic.*

Intuitively, Proposition 5 shows that if the patching domain $\mathcal{Z}'$ subsumes the input domain $\mathcal{Z}$, and the activation space $\mathcal{H}_G(\mathcal{Z}')$ is closed under concatenation, i.e., concatenating any two partial activations remains within $\mathcal{H}_G(\mathcal{Z}')$ — then the faithfulness predicate $\Phi$ is monotonic. This introduces a *new class of evaluation conditions* that are monotonic by construction, yielding stronger minimal circuits.

**Proposition 6.** *If the condition $\Phi(C, G)$ is set to validating whether $C$ is input-robust concerning $\langle f_G, \mathcal{Z} \rangle$ (Def. 1), and also patching-robust with respect to $\langle f_G, \mathcal{Z}' \rangle$ (Def. 2), then if $\mathcal{Z} \subseteq \mathcal{Z}'$ and $\mathcal{H}_G(\mathcal{Z}')$ is closed under concatenation, Algorithm 1 converges to a subset-minimal circuit.*

### 4.4 FROM SUBSET-MINIMAL CIRCUITS TO CARDINALLY-MINIMAL CIRCUITS

Although the monotonicity of $\Phi$ provides a stronger guarantee of subset-minimality, it still does not ensure convergence to the globally minimal circuit (i.e., a *cardinally*-minimal circuit). A naive approach to obtain such circuits is to enumerate all $C \subseteq G$, verify $\Phi(C, G)$, and choose the one with the lowest cardinality $|C|$, but this quickly becomes intractable even for modestly sized graphs.

**Exploiting circuit blocking-set duality for efficient approximation of cardinally-minimal circuits.** In this subsection, we leverage the idea that neural networks often contain small "circuit blocking-sets" — subgraphs $C' \subseteq G$ whose altered activations break the faithfulness of any circuit $C$ that excludes them. We prove a *duality* between circuits (under monotone faithfulness predicates) and these blocking-sets, enabling a new algorithmic construction that approximates — and sometimes exactly recovers — cardinally minimal circuits far more efficiently than naive search. Formally, for $f_G$ and $\Phi$, a *circuit blocking-set* is any $C' \subseteq G$ such that $\Phi(C \setminus C', G)$ fails for all $C \subseteq G$. This yields a duality grounded in a *minimum-hitting-set (MHS)* relation between circuits and blocking-sets:

**Proposition 7.** *Given some model $f_G$, and a monotonic predicate $\Phi$, the MHS of all circuit blocking-sets concerning $\Phi$ is a cardinally minimal circuit $C$ for which $\Phi(C, G)$ is true. Moreover, the MHS of all circuits $C \subseteq G$ for which $\Phi(C, G)$ is true, is a cardinally minimal blocking-set w.r.t $\Phi$.*

The definition of MHS, a classic NP-Complete problem is given in Appendix B.7. This duality is powerful because, despite NP-completeness, MHS can often be solved efficiently in practice with modern solvers such as MILP or MaxSAT (Ignatiev et al., 2019a). Hence, similar dualities have already been central to formal reasoning and provable explainability methods (Bacchus & Katsirelos, 2015; Ignatiev et al., 2019c; Bassan & Katz, 2023; Liffiton et al., 2016). With this duality theorem in hand, we can design an algorithm that often computes (or approximates) cardinally minimal circuits:

---

**Algorithm 2** Cardinally Minimal Circuit Approximation using MHS duality

---

1: **Input** model $f_G$, faithfulness predicate $\Phi$, $t_{\max} \in [|G|]$
2: BlockingSets $\leftarrow \emptyset$
3: **for** $t \leftarrow 1$ **to** $t_{\max}$ **do**
4:     $\mathcal{C}_t \leftarrow \{ S \subseteq G, \forall U \subseteq \text{BlockingSets} : |S| = t, U \not\subseteq S \}$
5:     **for all** $S \in \mathcal{C}_t$ **do**                               ▷ *parallelization*
6:         **if** $\neg\Phi(G \setminus S, G)$ **then**
7:             BlockingSets $\leftarrow$ BlockingSets $\cup \, S$
8:         **end if**
9:     **end for**
10:     $C \leftarrow \text{MHS}(\text{BlockingSets})$
11:     **if** $\Phi(C, G)$ **then return** $C$
12:     **end if**
13: **end for**

---

Algorithm 2 leverages Proposition 7 by iterating over blocking-sets *in parallel* and computing each set's MHS to obtain a circuit $C$. This establishes a lower bound on the cardinally minimal circuit and, through successive refinements, converges to the minimal one. While the number of blocking-set subsets may be excessive in the worst case, in practice their size is often tractable (see Section 5) yielding a low $t_{\max}$ and enabling more efficient — or closely approximate — computation of cardinally minimal circuits. This is formalized in the following claim:

**Proposition 8.** *Given a model $f_G$, and a monotonic predicate $\Phi$, Algorithm 2 computes a subset $C$ whose size is a* lower bound *to the cardinally minimal circuit for which $\Phi(C, G)$ is true. For a large enough $t_{max}$ value, the algorithm converges* exactly *to the cardinally minimal circuit.*

## 5 EXPERIMENTAL EVALUATION

**Experimental setup.** We evaluate our method on standard benchmarks from the neural network verification literature (Brix et al., 2024; 2023): (i) MNIST, (ii) CIFAR-10, (iii) GTSRB, and (iv) Taxi-iNet (a real-world dataset used in verification-based input-explainability studies (Wu et al., 2023a; Bassan et al., 2025a)) . For neural network verification, we use the state-of-the-art $\alpha, \beta$-CROWN verifier (Bak et al., 2021; Müller et al., 2022a; Brix et al., 2024); for MHS we use RC2 (Ignatiev et al., 2019a). For consistency, we evaluate our results on model architectures from prior verification studies, particularly the neural network verification competition (VNN-COMP) (Brix et al., 2024); full architectural details appear in Appendix D. To balance circuit discovery difficulty and human interpretability, we chose the circuit granularity level to be *neurons* for MNIST and *convolutional filters* for CIFAR-10, GTSRB, and TaxiNet. Both provable and sampling-based variants use the standard *logit-difference* metric (Conmy et al., 2023). Further details are in Appendices E, F and G.

### 5.1 CIRCUIT DISCOVERY WITH INPUT-ROBUSTNESS GUARANTEES

We begin by evaluating the *continuous input robustness* guarantees of our method. We compare two variants of Algorithm 1: (i) a standard sampling-based approach, where faithfulness is assessed by applying the logit-difference predicate with tolerance $\delta$ on sampled inputs, and (ii) a *provable* approach, which, via the siamese encoding of Section 3.1, certifies that the logit difference always remains below $\delta$ throughout the continuous input domain. For both methods, we report circuit size and robustness over the continuous input neighborhood across 100 batches (one circuit per batch). We set the input neighborhood using $\epsilon_p$ values aligned with VNN-COMP (Brix et al., 2024) (with variations in Appendix E) and adopt zero-patching in both settings. Results (Table 1) show that the provable method is slower, due to solving certification queries, yet achieves 100% robustness with *comparable circuit sizes*, whereas the sampling-based baseline attains substantially lower robustness. An illustrative example appears in Figure 4.

Table 1: Circuit results from Algorithm 1, where $\Phi$ is defined either by bounding logit differences under input sampling or by *verifying* the bound using the siamese encoding.

| Dataset | Sampling-based Circuit Discovery | | | Provably Input-Robust Circuit Discovery | | |
|---|---|---|---|---|---|---|
| | Time (s) | Size ($|C|$) | Robustness (%) | Time (s) | Size ($|C|$) | Robustness (%) |
| CIFAR-10 | 0.23 ±0.52 | 16.47 ±9.08 | **46.5** ±5.0 | 2970.85 ±874.23 | 19.18 ±10.16 | **100.0** ±0.0 |
| MNIST | 0.31 ±0.89 | 12.56 ±2.30 | **19.2** ±4.0 | 611.93 ±97.14 | 15.84 ±2.33 | **100.0** ±0.0 |
| GTSRB | 0.11 ±0.33 | 28.91 ±4.69 | **27.6** ±4.0 | 991.08 ±162.91 | 29.59 ±4.45 | **100.0** ±0.0 |
| TaxiNet | 0.01 ±0.00 | 5.77 ±0.80 | **9.5** ±3.0 | 180.00 ±40.39 | 6.82 ±0.46 | **100.0** ±0.0 |

### 5.2 CIRCUIT DISCOVERY WITH PATCHING-ROBUSTNESS GUARANTEES

Table 2: Circuit results from Algorithm 1, where $\Phi$ is defined either by bounding logit differences under zero patching, mean patching, or by *verifying* the bound using the siamese encoding.

| Dataset | Zero Patching | | | Mean Patching | | | Provably Patching-Robust Patching | | |
|---|---|---|---|---|---|---|---|---|---|
| | Time (s) | Size ($|C|$) | Rob. (%) | Time (s) | Size ($|C|$) | Rob. (%) | Time (s) | Size ($|C|$) | Rob. (%) |
| CIFAR-10 | 0.1 ± 0.3 | 65.1 ±3.0 | **46.4** ±6.0 | 0.0 ±0.0 | 64.1 ±3.6 | 33.3 ±5.7 | 5408.5 ±1091.0 | 65.6 ±1.6 | **100.0** |
| MNIST | 0.1 ±0.3 | 20.0 ±1.5 | **58.0** ±5.3 | 0.0 ±0.0 | 19.2 ±1.8 | 55.7 ±5.3 | 714.9 ±207.1 | 17.0 ±2.3 | **100.0** |
| GTSRB | 0.3 ±0.9 | 32.6 ±4.2 | **38.0** ±4.4 | 0.0 ±0.0 | 33.4 ±4.2 | 40.5 ±4.5 | 2907.2 ±721.7 | 34.3 ±4.1 | **100.0** |
| TaxiNet | 0.0 ±0.1 | 5.8 ±0.8 | **57.1** ±5.0 | 0.0 ±0.1 | 5.4 ±0.7 | 63.3 ±4.9 | 175.7 ±52.7 | 5.4 ±0.6 | **100.0** |

To assess patching robustness, we study three variants of Algorithm 1 enforcing a bounded logit difference under different patching schemes: (i) zero-patching, (ii) mean-patching, and (iii) a certified

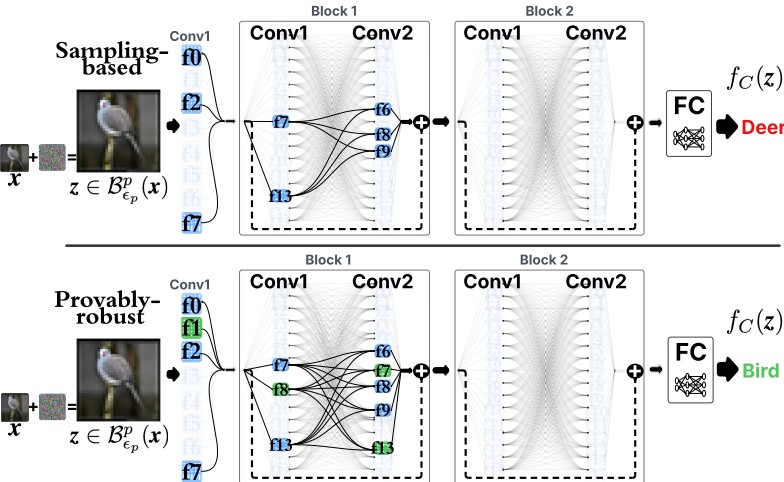

Figure 4: Examples of ResNet circuits at the filter level on CIFAR-10. Filters are numbered within each layer, with non-circuit filters in gray and residual connections shown as dashed lines. We compare circuits from the sampling-based discovery and the provably robust variant, highlighting components unique to the provable circuit in green.

variant that, using a siamese encoding (Section 3.2), verifies the bound uniformly over a continuous patching domain. Circuits found with zero or mean patching are then evaluated under the same continuous-domain criterion as the certified setting. Results appear in Table 2. Circuits found under standard patching (zero/mean) are sensitive to changes in the patching domain and yield low robustness, whereas the verified method certifies this property and achieves 100% robustness. Despite the higher computational cost (due to the reliance on verification), the provable method delivers much stronger robustness at comparable circuit sizes.

## 5.3 EXPLORING DIFFERENT MINIMALITY GUARANTEES OF CIRCUIT DISCOVERY

We experiment with the minimality guarantees from Sec. 4 and their connection to the robustness and patching guarantees of Sec. 3. For the $\Phi$ predicate, we certify *both* input- and patching robustness using a double-siamese encoding (Sec. 3.2), with environments $\mathcal{Z} \subseteq \mathcal{Z}'$, and run Alg. 1, 2, 4. Alg. 2 is run with $t_{\max} = 3$, restricting the contrastive blocking-set enumeration to sets of size at most three. Our experiments show that MHS size consistently lower-bounds circuit size, with no circuit falling *below* its MHS. In some runs, Alg. 1 circuits meet the bound exactly, and some MHS circuits are certified as faithful (i.e., satisfying both input and patching robustness), as shown in Fig. 5a.

We also note the *efficiency–circuit size trade-off* in Figure 5b: the quasi-minimal (Alg. 4) procedure terminates fastest but plateaus at larger sizes with weaker minimality; the iterative search (Alg. 1) achieves smaller sizes (stronger minimality) at higher runtime; and the MHS (Alg. 2) loop is slowest yet progressively approaches a cardinally minimal (optimal) circuit size.

## 6 RELATED WORK

**Circuit discovery.** Our work joins recent efforts on circuit discovery in MI (Olah et al., 2020; Elhage et al., 2021; Dunefsky et al., 2024), particularly those advancing *automated* algorithms (Conmy et al., 2023; Hsu et al., 2024). Other relevant avenues include metrics for circuit faithfulness (Marks et al., 2025; Hanna et al., 2024), differing patching strategies (Jafari et al., 2025; Syed et al., 2024; Haklay et al., 2025; Miller et al., 2024; Zhang & Nanda, 2024), minimality criteria (Chowdhary et al., 2025; Wang et al., 2023), and applications (Yao et al., 2024; Sharkey et al., 2025).

**Theoretical investigations of MI.** Several recent directions have examined other theoretical aspects of MI, often linked to circuit discovery. These include framing MI within a broader causal abstraction framework (Geiger et al., 2025; 2024), connecting it to distributed alignment search (DAS) (Wu

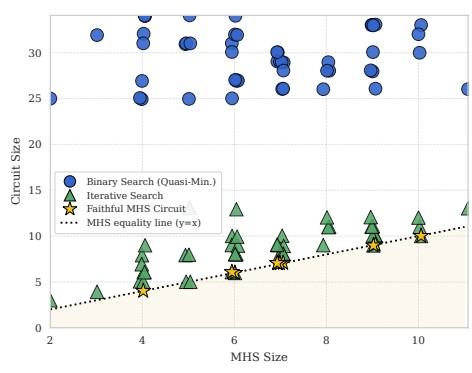
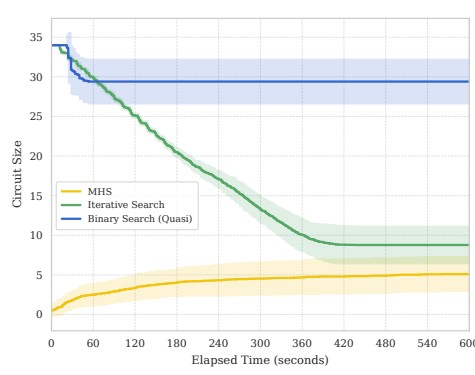

(a) Circuit size against MHS size                    (b) Circuit size over time

Figure 5: (a) Circuit size vs. MHS of blocking sets size, with the dashed equality line $y=x$ as the lower bound. (b) Convergence of circuit size over the first 10 minutes; shaded region shows deviation.

et al., 2023b; Sun et al., 2025); analyzing learned circuit logic through abstract interpretation from program analysis (Palumbo et al., 2025), proof theory (Miller et al., 2024; Wu et al., 2025), statistical identification (Méloux et al., 2025), and complexity theory (Adolfi et al., 2025).

**Neural network verification and formal explainability.** Our certification of robustness guarantees (input and patching) builds on the rapid progress of neural network verification (Brix et al., 2024; Wang et al., 2021; Zhou et al., 2024; Chiu et al., 2025; Müller et al., 2021; Singh et al., 2019). These advances have also been applied to certifying provable guarantees for input-based *explainability* notions (Wu et al., 2023a; Bassan et al., 2025a; Izza et al., 2024; Audemard et al., 2022b; La Malfa et al., 2021) (often termed formal explainable AI (Marques-Silva & Ignatiev, 2022)). Our work is the first to employ neural network verification based strategies for circuit discovery in mechanistic interpretability.

## 7 LIMITATIONS AND FUTURE WORK

A limitation of our framework, shared by all methods offering robustness guarantees over continuous domains, is its reliance on neural network verification queries. While current verification techniques remain limited for state-of-the-art models, they are advancing rapidly in scalability (Brix et al., 2024; Wang et al., 2021; Zhou et al., 2024). Our framework provides a novel integration of such tools to *mechanistic interpretability*, enabling circuit discovery with provable guarantees. Hence, as verification methods continue to scale, so will the reach of our approach, as our extensive experiments are grounded in $\alpha$-$\beta$-CROWN, the current leading verifier, and evaluated on standard benchmarks from the annual NN verification competition. Moreover, our novel *theoretical* results, covering guarantees over input domains, patching domains, and minimality, lay strong groundwork for future research on *provable* circuit discovery, including probabilistic and statistical forms of guarantees.

## 8 CONCLUSION

We introduce a framework for discovering circuits with provable guarantees, covering both (i) continuous *input*-domain robustness, (ii) continuous *patching*-domain robustness, and (iii) multiple forms of *minimality*. Central to our approach is the notion of *circuit monotonicity*, which reveals deep theoretical connections between input, patching, and minimality guarantees, and underpins the convergence of circuit discovery algorithms. Our experiments, which leverage recent advancements in neural network verification, confirm that our framework delivers substantially stronger guarantees than standard sampling-based approaches commonly used in circuit discovery. By bridging circuit discovery with neural network verification, this work takes a novel step toward designing *safer, more reliable circuits*, and lays new theoretical and algorithmic foundations for future research in provable circuit discovery.

## ACKNOWLEDGMENTS

This work was partially funded by the European Union (ERC, VeriDeL, 101112713). Views and opinions expressed are however those of the author(s) only and do not necessarily reflect those of the European Union or the European Research Council Executive Agency. Neither the European Union nor the granting authority can be held responsible for them. This research was additionally supported by a grant from the Israeli Science Foundation (grant number 558/24).

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

# Appendix

The appendix collects proofs, model specifications, and supplementary experimental results that support the main paper.

**Appendix A** contains additional background on neural network verification, circuit discovery, and patching.
**Appendix B** contains the complete proofs of all propositions.
**Appendix C** provides the pseudocode for the greedy quasi-minimality algorithm, together with a more detailed explanation of the toy example illustrating the minimality notion.
**Appendix D** provides specifications of the datasets and architectures used.
**Appendix E** provides additional details on the input-robustness experiment's methodology, verification setup, and evaluation.
**Appendix F** provides additional details on the patching-robustness experiment's methodology, verification setup, and evaluation.
**Appendix G** provides details on the minimality evaluation experiment's methodology, verification setup, and evaluation.
**Appendix H** provides the LLM usage discolsure.

## A    ADDITIONAL BACKGROUND ON NEURAL NETWORK VERIFICATION, CIRCUIT DISCOVERY, AND FORMAL XAI

**Neural Network Verification.**    Neural network verification provides *formal guarantees* about the behavior of a neural network $f_G$ over a continuous input region. Classic SMT/MILP-based approaches (Katz et al., 2017; Wu et al., 2024; Katz et al., 2019; Tjeng et al., 2017; Ehlers, 2017) encode ReLU networks and specifications as logical or mixed-integer constraints and offer exact guarantees, but scale only to small–medium models. Abstract-interpretation methods (Singh et al., 2019; Gehr et al., 2018; Ferrari et al., 2022; Müller et al., 2022b) propagate over-approximations layer by layer, giving fast but incomplete robustness certificates. A major advance came from linear-relaxation–based bound propagation, notably CROWN (Zhang et al., 2018) and follow-ups (Wang et al., 2021; Chiu et al., 2025; Zhou et al., 2024; Shi et al., 2025), which compute tight dual-based linear bounds and serve either as scalable incomplete verifiers or as strong relaxations inside exact search. Modern branch-and-bound (BaB) frameworks leverage these relaxations to achieve *complete* verification at scale, with $\alpha$-$\beta$-CROWN and related variants now dominating VNN-COMP (Brix et al., 2024) and handling million-parameter models. Recent progress includes tighter relaxations (e.g., SDP hybrids (Chiu et al., 2025)), cutting planes (Zhou et al., 2024), and support for non-ReLU nonlinearities (Shi et al., 2025). Verification today routinely certifies robustness for moderately large CNNs and ResNets, though major challenges remain for transformers, complex architectures, and richer temporal or relational specifications.

**Circuit discovery and patching.** An important step in circuit discovery is *patching*, which seeks to isolate the computational role of a hypothesized circuit by intervening on the activations outside it. In a typical setup, the model is run on a base input, and activations at selected non-circuit nodes are replaced — either with fixed baseline values (e.g., zero or mean activation) or with activations taken from a counterfactual input. If the model's output remains unchanged, the circuit is understood to be sufficient for the behavior; if it changes, this reveals a dependency on the patched components. Numerous patching protocols have been proposed, including activation replacement, path patching, and attention/head interventions (Jafari et al., 2025; Syed et al., 2024; Haklay et al., 2025; Miller et al., 2024; Zhang & Nanda, 2024; Nanda et al., 2023), all aiming to identify model components whose behavior is necessary or sufficient for a target computation. To the best of our knowledge, our method is the first to provably certify the stability of circuits under families of such patching interventions.

**Formal XAI.** Our work is also related to the line of research known as *formal explainable AI* (formal XAI) (Marques-Silva & Ignatiev, 2022), which studies input-based explanations with formal guarantees (Yu et al., 2023; Darwiche & Ji, 2022; Darwiche, 2023; Shih et al., 2018; Azzolin et al., 2025; Audemard et al., 2021). Since generating explanations with formal guarantees is often computationally intractable (Adolfi et al., 2025; Barceló et al., 2020; Arenas et al., 2021; Bassan et al., 2026b; Marzouk et al., 2025a;b; Marzouk & De La Higuera, 2024; Amir et al., 2024; Bassan et al.,

2024; Blanc et al., 2021; 2022), much of this literature has focused on simpler model classes (Marques-Silva & Ignatiev, 2023), including decision trees (Bounia, 2025; Arenas et al., 2022; Bounia, 2024), linear models (Marques-Silva et al., 2020), additive models (Bassan et al., 2025e; 2026a), monotonic models (Marques-Silva et al., 2021; Harzli et al., 2023), and tree ensembles (Audemard et al., 2023; 2022a; Ignatiev et al., 2022; Bassan et al., 2025b; Boumazouza et al., 2021). Closer to our setting, several works have extended these guarantees to neural networks (Malfa et al., 2021; Bassan et al., 2023; Wu et al., 2023a; Bassan et al., 2025a; Boumazouza et al., 2026; Labbaf et al., 2025; De Palma et al., 2025; Fel et al., 2023; Soria et al., 2025).

A central concept in formal explainability is that of a *sufficient reason* (Darwiche & Hirth, 2020; Bassan et al., 2025c;d; Barceló et al., 2020), also referred to as an *abductive explanation* (Ignatiev et al., 2019b; Izza et al., 2023). These notions are closely related to stability-based feature attribution frameworks (Xue et al., 2023; Jin et al., 2025; Anani et al., 2025). Informally, a sufficient reason resembles a circuit in that fixing it preserves the model's prediction. Various minimality notions have been studied in this context, including subset-minimal explanations (Ignatiev et al., 2019b), as well as cardinally-minimal explanations that are computed via a similar minimal hitting set (MHS) duality (Ignatiev et al., 2019b; Bassan & Katz, 2023) that was studied here. However, our work addresses a fundamentally different task: discovering *circuits* rather than input subsets. Moving from subsets of features to subcomponents of the model introduces several new challenges, including dual encodings of the model for certification, the need to handle patching operations, and the loss of monotonicity under certain patching schemes, which complicates convergence to minimal circuits. To the best of our knowledge, this is the first work to leverage ideas from formal explainability within mechanistic interpretability, and in particular for the task of circuit discovery.

Lastly, another partially relevant line of work has incorporated neural network verification to activation-pattern specifications (NAPs) (Geng et al., 2023), which encode active/inactive neuron states and induce input regions beyond local perturbation balls. Recent work (Geng et al., 2024) further aims to learn minimal NAPs by removing redundant neuron states while preserving correctness. These also differ from our circuits which compute actual subgraphs of neural networks.

# B   PROOFS OF MAIN RESULTS

This appendix presents the proofs of the main propositions stated in the main paper.

## B.1   PROOF OF PROPOSITION 1

**Proposition 1.** *Given any model $f_G$ and faithfulness predicate $\Phi$, Algorithm 1 visits a* quasi-minimal *circuit and Algorithm 3 converges to a* locally-minimal *circuit $C$, both with respect to $\Phi$.*

*Proof.* We must establish three points: (i) *Faithfulness:* The final circuit $C$ returned by Algorithms 1 and 3 is indeed faithful, i.e., the predicate $\Phi(C, G)$ holds true. (ii) Algorithm 1 visits a quasi-minimal circuit during its execution, i.e., there exists an iteration $t$ where the corresponding circuit $C_t$ contains a component $i$ such that $\Phi(C_t \setminus \{i\}, G)$ is false. (iii) Algorithm 3 achieves local minimality: In the returned circuit $C$, removing any single component $i \in C$ breaks faithfulness; that is, for every $i \in C$, $\Phi(C \setminus \{i\}, G)$ is false.

In the first part of the proof, both Algorithms 1 and 3 start with $C \leftarrow G$. Since the full model graph $G$ is, by definition, faithful, $\Phi(G, G)$ holds true. At each step, the algorithms check $\Phi(C \setminus \{i\}, G)$ for the current component $i$ and update $C \leftarrow C \setminus \{i\}$ only if this predicate remains true. Hence, the invariant is preserved throughout the execution, and the final circuit $C$ still satisfies $\Phi(C, G)$.

For the second part, we prove that Algorithm 1 visits a quasi-minimal circuit during its execution. We assume that the empty circuit is unfaithful, i.e., $\neg\Phi(\emptyset, G)$. Let $C_t$ denote the circuit maintained by the algorithm at iteration $t$. Since the algorithm maintains the invariant $\Phi(C_t, G)$ and removes components only when faithfulness is preserved, it cannot reach the empty circuit. Hence, not all components can be removed while maintaining $\Phi$. Therefore, there exists a first iteration $t$ at which the algorithm considers some component $i \in C_t$ and does not remove it because $\Phi(C_t \setminus \{i\}, G)$ is false. Thus, $C_t$ contains a component whose removal violates faithfulness and is, by definition, quasi-minimal with respect to $\Phi$. This establishes that Algorithm 1 visits a quasi-minimal circuit during its execution.

For the third part of the proof, note that Algorithm 3 continues to iterate over the circuit components until a full pass occurs with no further changes to the circuit $C$. This termination condition implies that for any remaining $i \in C$, the predicate $\Phi(C \setminus \{i\}, G)$ must have been false during the final iteration. Otherwise, if $\Phi(C \setminus \{i\}, G)$ were true for any $i \in C$, the component would have been removed, and the algorithm would have proceeded to another iteration. Therefore, upon termination, every $i \in C$ is essential for maintaining the faithfulness predicate, establishing that $C$ is locally minimal with respect to $\Phi$. This completes the proof.

□

### B.2 PROOF OF PROPOSITION 2

**Proposition 2.** *While Algorithm 4 converges to a quasi-minimal circuit and performs $\mathcal{O}(\log |G|)$ evaluations of $\Phi(C, G)$ (Adolfi et al., 2025), Algorithm 1 visits a quasi-minimal circuit and performs $\mathcal{O}(|G|)$ such evaluations, and Algorithm 3 converges to a locally-minimal circuit and performs $\mathcal{O}(|G|^2)$ evaluations.*

*Proof.* We provide the proof for each algorithm:

*(Algorithm 1).* By Proposition 1, Algorithm 1 visits a quasi-minimal circuit during its execution. Observe that the algorithm 1 processes each component of $G$ individually and exactly once. Since it tests $\Phi$ upon the removal of each of these $|G|$ components, the procedure carries out $\mathcal{O}(|G|)$ predicate evaluations in total.

*(Algorithm 3).* By Proposition 1, Algorithm 3 converges to a locally minimal circuit. For the runtime analysis, observe that Algorithm 3 processes each component of $G$ and continues to do so until convergence. In each pass, it tests the predicate $\Phi$ upon the removal of every remaining component. In the worst case, when only a single component is removed in each pass, the total number of predicate evaluations is $|G| + (|G| - 1) + (|G| - 2) + \cdots + 1 = \mathcal{O}(|G|^2)$. Thus, the procedure performs $\mathcal{O}(|G|^2)$ evaluations of $\Phi(C, G)$ in total.

*(Algorithm 4).* The quasi-minimality algorithm 4 and its runtime were established in Adolfi et al. (2025). For completeness, we restate the argument regarding the number of evaluations: the algorithm halves the candidate index range at each step, requiring at most $\lceil \log |G| \rceil$ predicate evaluations before termination. Thus, its complexity is $\mathcal{O}(\log |G|)$.

□

### B.3 PROOF OF PROPOSITION 3

**Proposition 3.** *There exist infinitely-many number of configurations of $f_G$, and $\Phi$, for which Algorithms 1, 3 and 4 do not converge to a subset-minimal circuit $C$ concerning $\Phi$.*

*Proof.* Consider a small nonlinear counterexample network $f_G : \mathbb{R} \to \mathbb{R}$, with underlying structure $G$ and node set $V_G := \{v_1, v_2, v_3, v_4\}$. The network is defined over a one-dimensional input $x$, with three hidden nodes $v_1, v_2, v_3$ and an output node $v_4$.

Consider a one-dimensional input $x$, three hidden units $v_1, v_2, v_3$, and an output neuron $v_4$:

$$v_1(x) := \mathrm{ReLU}(x), \qquad v_2(x) := \mathrm{ReLU}(x), \qquad v_3(x) := x, \qquad v_4(u, v, w) := u - v + w \quad (3)$$

Therefore, the following holds:

$$f_G(x) := v_4\big(v_1(x), v_2(x), v_3(x)\big) = v_1(x) - v_2(x) + v_3(x) = x \qquad (4)$$

For any subset $C \subseteq \{v_1, v_2, v_3\}$, define $f_C$ by applying *zero-patching* to all hidden units outside $C$. We take the faithfulness predicate to be *strong equality*, i.e., corresponding to $\delta = 0$ and $\epsilon_p$ given by the $\ell_\infty$ norm over the domain $[0, 1]$. In particular:

$$\Phi(C, G) \iff \forall x \in [0, 1], \ f_C(x) = f_G(x) \qquad (5)$$

*Iterative (Algorithms 1 and 3).* We consider the standard traversal order over these components from end to start (i.e., reverse topological order): $(v_4, v_3, v_2, v_1)$. We note that removing $v_4$ (i.e., zero-patching it) directly breaks faithfulness. Moreover, no other single-component removal preserves $\Phi$: removing any hidden unit breaks the equality.

$$
\begin{aligned}
f_{G \setminus \{v_3\}}(x) &= \text{ReLU}(x) - \text{ReLU}(x) = 0 \neq x \text{ for } x > 0, \\
f_{G \setminus \{v_2\}}(x) &= \text{ReLU}(x) + x \neq x \text{ for } x > 0, \\
f_{G \setminus \{v_1\}}(x) &= -\text{ReLU}(x) + x \neq x \text{ for } x > 0
\end{aligned} \tag{6}
$$

Hence, Algorithm 1 and Algorithm 3 halt at $C = G$ (locally minimal). Yet a strict subset is faithful: removing the pair $\{v_1, v_2\}$ yields $f_{G \setminus \{v_1, v_2\}}(x) = v_3(x) = x$, so $\Phi(G \setminus \{v_1, v_2\}, G)$ holds. Therefore, $G$ is not subset-minimal.

*Quasi (Algorithm 4).* With a valid order $(v_1, v_3, v_2, v_4)$, Algorithm 4 first tests the prefix removal $\{v_1, v_3\}$, leaving $f_G(x) = -\text{ReLU}(x) \neq x$, then tests $\{v_1\}$, leaving $f_G(x) = -\text{ReLU}(x) + x \neq x$; it never considers the pair $\{v_1, v_2\}$ and so returns $C = G$, which is not subset-minimal, as we previously showed in Equation 6.

Since this holds for every $m \in \mathbb{N}$, we obtain infinitely many configurations where both algorithms fail to return a subset-minimal circuit.

*Infinite family.* For any $m \geq 1$, add pairs $(p_i, q_i)$ with $p_i(x) = q_i(x) = \text{ReLU}(x)$ and set

$$
g_v(x) = v_3(x) + \sum_{i=1}^{m} \big( p_i(x) - q_i(x) \big).
$$

Then $f_G(x) = x$ on $[0, 1]$, any single removal (of $v_3$, some $p_j$, or some $q_j$) breaks equality, but removing a whole pair $\{p_j, q_j\}$ preserves it. Under the ordering $(p_1, \ldots, p_m, v_3, q_1, \ldots, q_m, v_4)$, the iterative greedy Algorithms 1 and 3 and the quasi-minimal Algorithm 4 both return $G$. Thus, there are infinitely many such configurations.

$\square$

### B.4 PROOF OF PROPOSITION 4

**Proposition 4.** *If $\Phi$ is monotonic, then for any model $f_G$, Algorithm 1 converges to a subset-minimal circuit $C$ concerning $\Phi$.*

*Proof.* By definition, establishing subset-minimality of the circuit $C$ requires showing: (i) that $C$ is faithful, i.e., $\Phi(C, G)$ holds, and (ii) that no proper subgraph $C' \subseteq C$ also satisfies $\Phi(C', G)$.

For the first part of the proof, we follow the same faithfulness-preservation argument used in the proof of Proposition 1. Algorithm 1 begins with the full model $G$, which is initially faithful to itself ($\Phi(G, G)$ holds). It then iterates through components in a fixed order (e.g., reverse topological order). At each step, the algorithm maintains a current circuit $C_t$; a component $i$ is removed only if its removal preserves faithfulness, i.e., if $\Phi(C_t \setminus \{i\}, G)$ holds. Consequently, faithfulness is maintained as a loop invariant throughout the execution. In particular, upon termination, the resulting circuit $C$ satisfies $\Phi(C, G)$.

For the second part of the proof, assume towards contradiction that there exists some $C' \subsetneq C$ for which it holds that $\Phi(C', G)$ is true. Now, pick any $c^\star \in C \setminus C'$. Let $C_t$ be the circuit at iteration $t$ where this iteration marks the step over which the algorithm has evaluated whether to add $c^\star$ to the circuit. Because $c^\star$ is present in the *final* returned circuit $C$, then by induction it was not removed at step $t$ when considered, and hence it must hold that $\Phi(C_t \setminus \{c^\star\}, G)$ is false. On the other hand, we have the following inclusions:

$$
C' \subseteq C \setminus \{c^\star\} \subseteq C_t \setminus \{c^\star\}. \tag{7}
$$

And so by the very definition of the monotonicity of $C$ with respect to $\Phi$, we obtain that:

$$
\Phi(C', G) \implies \Phi(C_t \setminus \{c^\star\}, G) \tag{8}
$$

This contradicts the fact that we have derived that $\Phi(C_t \setminus \{c^\star\}, G)$ must be false from the algorithm's progression. Hence, we have obtained that for any subset $C' \subsetneq C$ it holds that $\Phi(C', G)$ is false, and hence $C$ is subset-minimal with respect to $\Phi$.

$\square$

### B.5 PROOF OF PROPOSITION 5

**Proposition 5.** *Let $\Phi(C, G)$ denote validating whether $C$ is input-robust concerning $\langle f_G, \mathcal{Z} \rangle$ (Def. 1), and simultaneously patching-robust concerning $\langle f_G, \mathcal{Z}' \rangle$ (Def. 2). Then if $\mathcal{Z} \subseteq \mathcal{Z}'$, and $\mathcal{H}_G(\mathcal{Z}')$ is closed under concatenation, $\Phi$ is monotonic.*

*Proof.* We begin by formally stating the condition of an activation space being closed under concatenation:

**Definition 1.** *We say that an activation space $\mathcal{H}_G(\mathcal{Z})$ of some model $f_G$ and domain $\mathcal{Z} \subseteq \mathbb{R}^n$ is closed under concatenation iff for any two partial activations $\alpha, \alpha'$, where $\alpha \in \mathcal{H}_C(\mathcal{Z})$ is an activation over the circuit $C \subseteq G$, and $\alpha' \in \mathcal{H}_{C'}(\mathcal{Z})$ is an activation over the circuit $C' \subseteq G$, then it holds that $\alpha \cup \alpha' \in \mathcal{H}_{C \cup C'}(\mathcal{Z})$.*

Now, let there be some $C \subseteq C' \subseteq G$, for which $C' := C \sqcup \{c'_1, \ldots, c'_t\}$. We assume that the predicate $\Phi$ is defined as validating whether some circuit $C$ is input-robust concerning $\langle f_G, \mathcal{Z} \rangle$, and also patching-robust with respect to $\langle f_G, \mathcal{Z}' \rangle$. In other words, this implies that $\Phi(C, G)$ holds if and only if:

$$\forall \mathbf{z} \in \mathcal{Z} := \bigcup_{j=1}^{k} \mathcal{B}_{\epsilon_p}^p(\mathbf{x}_j), \quad \forall \alpha \in \mathcal{H}_{\overline{C}}(\mathcal{Z}') : \quad \left\| f_C(\mathbf{z} \mid \overline{C} = \alpha) - f_G(\mathbf{z}) \right\|_p \leq \delta \tag{9}$$

We note that the following notation is equivalent to the following:

$$\begin{aligned} \max_{\mathbf{z} \in \mathcal{Z}, \alpha \in \mathcal{H}_{\overline{C}}(\mathcal{Z}')} \quad & \left\| f_C(\mathbf{z} \mid \overline{C} = \alpha) - f_G(\mathbf{z}) \right\|_p \leq \delta \iff \\ \max_{\mathbf{z} \in \mathcal{Z}, \mathbf{z}' \in \mathcal{Z}'} \quad & \left\| f_C(\mathbf{z} \mid \overline{C} = \mathcal{H}_{\overline{C}}(\mathbf{z}')) - f_G(\mathbf{z}) \right\|_p \leq \delta \end{aligned} \tag{10}$$

where we use the notation $\mathcal{H}_{\overline{C}}(\mathbf{z}')$ to denote the *specific* activation over $\overline{C}$ when computing $f_G(\mathbf{z}')$ for some $\mathbf{z}' \in \mathbb{R}^n$.

Since our goal is to prove that $\Phi$ is monotonic, it suffices to show that $C'$ also satisfies the above conditions. An equivalent formulation is to verify the condition for any $C' := C \sqcup \{c'_i\}$, i.e., for supersets that differ from $C$ by a single element rather than an arbitrary subset. This is valid because adding subsets inductively, one element at a time, is equivalent to adding the entire subset at once. We therefore proceed under this formulation and assume, for contradiction, that the conditions fail to hold for $C'$. In other words, we assume the following:

$$\begin{aligned} \exists \mathbf{z} \in \mathcal{Z}, \quad \exists \alpha \in \mathcal{H}_{\overline{C'}}(\mathcal{Z}') : \quad & \left\| f_{C'}(\mathbf{z} \mid \overline{C'} = \alpha) - f_G(\mathbf{z}) \right\|_p > \delta \iff \\ \max_{\mathbf{z} \in \mathcal{Z}, \alpha \in \mathcal{H}_{\overline{C'}}(\mathcal{Z}')} \quad & \left\| f_{C'}(\mathbf{z} \mid \overline{C'} = \alpha) - f_G(\mathbf{z}) \right\|_p > \delta \iff \\ \max_{\mathbf{z} \in \mathcal{Z}, \mathbf{z}' \in \mathcal{Z}'} \quad & \left\| f_{C'}(\mathbf{z} \mid \overline{C'} = \mathcal{H}_{\overline{C'}}(\mathbf{z}')) - f_G(\mathbf{z}) \right\|_p > \delta \end{aligned} \tag{11}$$

This is also equivalent to stating that:

$$\max_{\mathbf{z} \in \mathcal{Z}, \mathbf{z}' \in \mathcal{Z}'} \quad \left\| f_{C \sqcup \{c'_i\}}(\mathbf{z} \mid \overline{C \sqcup \{c'_i\}} = \mathcal{H}_{\overline{C \sqcup \{c'_i\}}}(\mathbf{z}')) - f_G(\mathbf{z}) \right\|_p > \delta \tag{12}$$

Let us denote by $\mathbb{S} \subseteq \mathbb{R}$ the set of all values that are feasible to obtain by $f_C(\mathbf{z} \mid \overline{C} = \mathcal{H}_{\overline{C}}(\mathbf{z}'))$ and by $\mathbb{S} \subseteq \mathbb{R}$ all values that are feasible by $f_C(\mathbf{z} \mid \overline{C'} = \mathcal{H}_{\overline{C'}}(\mathbf{z}'))$. More precisely:

$$\mathbb{S} := \{f_C(\mathbf{z} \,|\, \overline{C} = \mathcal{H}_{\overline{C}}(\mathbf{z}')) : \mathbf{z} \in \mathcal{Z}, \mathbf{z}' \in \mathcal{Z}'\}, \wedge \mathbb{S}' := \{f_{C'}(\mathbf{z} \,|\, \overline{C'} = \mathcal{H}_{\overline{C'}}(\mathbf{z}')) : \mathbf{z} \in \mathcal{Z}, \mathbf{z}' \in \mathcal{Z}'\} \tag{13}$$

For finalizing the proof of the proposition, we will make use of the following Lemma:

**Lemma 1.** *Given the predefined $f_G$, the circuits $C \subseteq C' \subseteq G$, and the aforementioned notations of $\mathbb{S}$ and $\mathbb{S}'$, then it holds that $\mathbb{S}' \subseteq \mathbb{S}$.*

*Proof.* We first note that by definition:

$$\mathbb{S}' := \{f_{C'}(\mathbf{z} \,|\, \overline{C'} = \mathcal{H}_{\overline{C'}}(\mathbf{z}')) : \mathbf{z} \in \mathcal{Z}, \mathbf{z}' \in \mathcal{Z}'\} = \\ \{f_{C \sqcup \{c'_i\}}(\mathbf{z} \,|\, \overline{C \sqcup \{c'_i\}} = \mathcal{H}_{\overline{C \sqcup \{c'_i\}}}(\mathbf{z}')) : \mathbf{z} \in \mathcal{Z}, \mathbf{z}' \in \mathcal{Z}'\} \tag{14}$$

We also note that for any $\mathbf{z} \in \mathbb{R}^n$ it holds, by definition, that:

$$f_C(\mathbf{z} \,|\, \overline{C} = \mathcal{H}_{\overline{C}}(\mathbf{z}')) = f_\emptyset(\mathbf{z} \,|\, \overline{C} = \mathcal{H}_{\overline{C}}(\mathbf{z}'), C = \mathcal{H}_C(\mathbf{z})) \tag{15}$$

The notation $f_\emptyset(\mathbf{z} \,|\, \overline{C} = \mathcal{H}_{\overline{C}}(\mathbf{z}'), C = \mathcal{H}_C(\mathbf{z}))$ simply means that we fix the activations of $C$ to $\mathcal{H}_C(\mathbf{z})$ and those of $\overline{C}$ to $\mathcal{H}_{\overline{C}}(\mathbf{z}')$. From the same manipulation of notation, we can get that:

$$f_{C \sqcup \{c'_i\}}(\mathbf{z} \,|\, \overline{C \sqcup \{c'_i\}} = \mathcal{H}_{\overline{C \sqcup \{c'_i\}}}(\mathbf{z}')) = \\ f_\emptyset(\mathbf{z} \,|\, \overline{C \sqcup \{c'_i\}} = \mathcal{H}_{\overline{C \sqcup \{c'_i\}}}(\mathbf{z}'), C \sqcup \{i\} = \mathcal{H}_{C \sqcup \{c'_i\}}(\mathbf{z})) = \\ f_\emptyset(\mathbf{z} \,|\, \overline{C \sqcup \{c'_i\}} = \mathcal{H}_{\overline{C \sqcup \{c'_i\}}}(\mathbf{z}'), C = \mathcal{H}_C(\mathbf{z}), \{c'_i\} = \mathcal{H}_{c'_i}(\mathbf{z})) = \\ f_\emptyset(\mathbf{z} \,|\, \overline{C} \setminus \{c'_i\} = \mathcal{H}_{\overline{C} \setminus \{c'_i\}}(\mathbf{z}'), C = \mathcal{H}_C(\mathbf{z}), \{c'_i\} = \mathcal{H}_{c'_i}(\mathbf{z})) \tag{16}$$

To prove that $\mathbb{S}' \subseteq \mathbb{S}$, let us take some $\mathbf{z}_0, \mathbf{z}'_0 \in \mathcal{Z}$. We will now prove that for any such choice of $\mathbf{z}_0, \mathbf{z}'_0$ then the following holds:

$$f_{C'}(\mathbf{z}_0 \,|\, \overline{C'} = \mathcal{H}_{\overline{C'}}(\mathbf{z}'_0)) \subseteq \mathbb{S} \tag{17}$$

Since $\mathbf{z}'_0 \in \mathcal{Z}'$, $\mathbf{z}_0 \in \mathcal{Z} \subseteq \mathcal{Z}'$, and $\mathcal{H}_G(\mathcal{Z}')$ is closed under concatenation, then by definition it holds that fixing the activations of $\overline{C} \setminus \{c'_i\}$ to $\mathcal{H}_{\overline{C} \setminus \{c'_i\}}(\mathbf{z}'_0) \in \mathcal{H}_{\overline{C} \setminus \{c'_i\}}(\mathcal{Z}')$ and those of $\{c'_i\}$ to $\mathcal{H}_{\{c'_i\}}(\mathbf{z}_0) \in \mathcal{H}_{\{c'_i\}}(\mathcal{Z}')$ yields an activation $\alpha \in \mathcal{H}_{\overline{C} \setminus \{c'_i\} \cup \{c'_i\}}(\mathcal{Z}') = \mathcal{H}_{\overline{C}}(\mathcal{Z}')$.

Hence, we arrive at:

$$f_{C'}(\mathbf{z}_0 \,|\, \overline{C'} = \mathcal{H}_{\overline{C'}}(\mathbf{z}'_0)) = \\ f_{C \sqcup \{c'_i\}}(\mathbf{z}_0 \,|\, \overline{C \sqcup \{c'_i\}} = \mathcal{H}_{\overline{C \sqcup \{c'_i\}}}(\mathbf{z}'_0)) = \\ f_\emptyset(\mathbf{z}_0 \,|\, \overline{C} \setminus \{c'_i\} = \mathcal{H}_{\overline{C} \setminus \{c'_i\}}(\mathbf{z}'_0), C = \mathcal{H}_C(\mathbf{z}_0), \{c'_i\} = \mathcal{H}_{c'_i}(\mathbf{z}_0)) = \\ f_\emptyset(\mathbf{z}_0 \,|\, \overline{C} = \alpha, C = \mathcal{H}_C(\mathbf{z}_0)) \tag{18}$$

Since we have shown that $\alpha \in \mathcal{H}_{\overline{C}}(\mathcal{Z}')$ and since $\mathcal{H}_C(\mathbf{z}_0) \in \mathcal{H}_C(\mathcal{Z})$ then we have that:

$$f_{C'}(\mathbf{z}_0 \,|\, \overline{C'} = \mathcal{H}_{\overline{C'}}(\mathbf{z}'_0)) = \\ f_\emptyset(\mathbf{z}_0 \,|\, \overline{C} = \alpha, C = \mathcal{H}_C(\mathbf{z}_0)) \in \\ \{f_C(\mathbf{z} \,|\, \overline{C} = \mathcal{H}_{\overline{C}}(\mathbf{z}')) : \mathbf{z} \in \mathcal{Z}, \mathbf{z}' \in \mathcal{Z}'\} = \mathbb{S} \tag{19}$$

This establishes that $\mathbb{S}' \subseteq \mathbb{S}$, and hence concludes the proof of the lemma.

$\square$

Now to finalize the proof of the proposition, we recall that we have shown that the following holds (and can now rewrite this expression given our new definition of $\mathbb{S}$):

$$\max_{\mathbf{z}\in\mathcal{Z},\mathbf{z}'\in\mathcal{Z}'} \quad \big\| f_C(\mathbf{z}\,|\,\overline{C}=\mathcal{H}_{\overline{C}}(\mathbf{z}')) - f_G(\mathbf{z}) \big\|_p \leq \delta \iff$$
$$\max_{\mathbf{z}\in\mathcal{Z},\mathbf{y}\in\mathbb{S}} \quad \big\| \mathbf{y} - f_G(\mathbf{z}) \big\|_p \leq \delta \tag{20}$$

We have also assumed towards contradiction that the following holds, and we can similarly further rewrite this term given our new definition of $\mathbb{S}'$:

$$\max_{\mathbf{z}\in\mathcal{Z},\mathbf{z}'\in\mathcal{Z}'} \quad \big\| f_{C'}(\mathbf{z}\,|\,\overline{C'}=\mathcal{H}_{\overline{C'}}(\mathbf{z}')) - f_G(\mathbf{z}) \big\|_p > \delta \iff$$
$$\max_{\mathbf{z}\in\mathcal{Z},\mathbf{y}\in\mathbb{S}'} \quad \big\| \mathbf{y} - f_G(\mathbf{z}) \big\|_p > \delta \tag{21}$$

However, since we have proven in Lemma 1 that $\mathbb{S}' \subseteq \mathbb{S}$ then we know that:

$$\max_{\mathbf{z}\in\mathcal{Z},\mathbf{y}\in\mathbb{S}} \quad \big\| \mathbf{y} - f_G(\mathbf{z}) \big\|_p \geq \max_{\mathbf{z}\in\mathcal{Z},\mathbf{y}\in\mathbb{S}'} \quad \big\| \mathbf{y} - f_G(\mathbf{z}) \big\|_p \tag{22}$$

which stands in contradiction to equations 20 and 21, hence implying the monotonicity of $\Phi$, and concluding the proof of the proposition.

$\square$

### B.6 PROOF OF PROPOSITION 6

**Proposition 6.** *If the condition $\Phi(C, G)$ is set to validating whether $C$ is input-robust concerning $\langle f_G, \mathcal{Z}\rangle$ (Def. 1), and also patching-robust with respect to $\langle f_G, \mathcal{Z}'\rangle$ (Def. 2), then if $\mathcal{Z} \subseteq \mathcal{Z}'$ and $\mathcal{H}_G(\mathcal{Z}')$ is closed under concatenation, Algorithm 1 converges to a subset-minimal circuit.*

*Proof.* The claim follows directly from Propositions 4 and 5. Since $\mathcal{Z} \subseteq \mathcal{Z}'$, Proposition 5 implies that $\Phi(C, G)$ is *monotonic*. By Proposition 4, it follows that Algorithm 1 converges to a *subset-minimal* circuit $C$ with respect to $\Phi$.

$\square$

### B.7 PROOF OF PROPOSITION 7

**Proposition 7.** *Given some model $f_G$, and a monotonic predicate $\Phi$, the MHS of all circuit blocking-sets concerning $\Phi$ is a cardinally minimal circuit $C$ for which $\Phi(C, G)$ is true. Moreover, the MHS of all circuits $C \subseteq G$ for which $\Phi(C, G)$ is true, is a cardinally minimal blocking-set w.r.t $\Phi$.*

*Proof.* Prior to the proof of Proposition 7, which establishes the connection between Minimum Hitting Sets (MHS) and cardinal minimality, we first recall the definition of MHS:

**Definition 2** (Minimum Hitting Set (MHS)). *Given a collection $\mathcal{S}$ of sets over a universe $U$, a hitting set $H \subseteq U$ for $\mathcal{S}$ is a set such that*

$$\forall S \in \mathcal{S}, \quad H \cap S \neq \emptyset.$$

*A hitting set $H$ is called* minimal *if no subset of $H$ is a hitting set, and* minimum *if it has the smallest possible cardinality among all hitting sets.*

We also show the following equivalence concerning the set of blocking sets.

**Lemma 2.** *Under a monotone predicate $\Phi$, the set of circuit blocking sets with respect to $G$ coincides exactly with the sets whose removal breaks faithfulness of the full model, i.e., $C' \subseteq G$ is a blocking set if and only if $\neg\Phi(G \setminus C', G)$.*

*Proof.* We prove both directions, starting with $\Rightarrow$. Let $C'$ be a blocking set. By the definition of a blocking set (Section 4.4), $\Phi(C \setminus C', G)$ fails for all $C \subseteq G$, trivially including $C = G$. Therefore, $\neg\Phi(G \setminus C', G)$. For the reverse direction $\Leftarrow$, consider a set $C' \subseteq G$ whose removal from $G$, breaks the faithfulness of the full model $G$, namely $\neg\Phi(G \setminus C', G)$. Assume toward a contradiction that $C'$

is not a blocking set. Then there exists some $C \subseteq G$ for which $\Phi(C \setminus C', G)$ holds. By monotonicity of $\Phi$, since $C \subseteq G$, we have $\Phi(C \setminus C', G) \Rightarrow \Phi(G \setminus C', G)$, contradicting our assumption.

This completes the proof, establishing that under monotonicity, the blocking sets are exactly the sets whose removal breaks faithfulness of the full model $G$.

$\square$

We now move to prove that the MHS of blocking-sets is a cardinally minimal faithful circuit. Let $C$ be a minimum hitting set (MHS) of the set of blocking-sets $\mathcal{B}$.

We now move to prove Proposition 7, establishing that the minimum hitting set (MHS) of the blocking sets is a cardinally minimal faithful circuit. Let $\mathcal{B}$ denote the set of blocking sets, which by Lemma 2, under the monotonicity assumption, coincides with the sets whose removal breaks faithfulness of the full model $G$. Let $C$ be a minimum hitting set of $\mathcal{B}$.

Assume towards contradiction that $\neg\Phi(C, G)$. Set $B^\star := G \setminus C$. Then $\Phi(G \setminus B^\star, G) = \Phi(C, G)$ is false, so $B^\star \in \mathcal{B}$. Yet by definition $C \cap B^\star = \emptyset$, contradicting that $C$ hits every set in $\mathcal{B}$. Hence $\Phi(C, G)$ holds.

We now move forward to prove minimality. Assume there exists $C' \subseteq G$ with $\Phi(C', G)$ and $|C'| < |C|$. We claim $C'$ is also a hitting set of $\mathcal{B}$, contradicting the minimality of $C$ as an MHS. Indeed, if some $B \in \mathcal{B}$ satisfied $C' \cap B = \emptyset$, then $C' \subseteq G \setminus B$, and by monotonicity of $\Phi$ we would have $\Phi(G \setminus B, G)$, contradicting $B \in \mathcal{B}$. Hence $C'$ hits all of $\mathcal{B}$, contradicting that $C$ is an MHS.

For the second part of the proof, let $\mathcal{C} := \{ C \subseteq G : \Phi(C, G) \}$ and let $B$ be a minimum hitting set of $\mathcal{C}$. Assume towards contradiction that it is not, namely $\Phi(G \setminus B, G)$ holds. Then $C^\star := G \setminus B$ is a faithful circuit, implying $C^\star \in \mathcal{C}$. Yet by definition $C^\star \cap B = \emptyset$, contradicting that $B$ hits every set in $\mathcal{C}$.

Finally, assume towards contradiction that there exists a blocking-set $B'$ with $|B'| \leq |B|$. Let $C \in \mathcal{C}$. If $C \cap B' = \emptyset$, then $C \subseteq G \setminus B'$, and by monotonicity of $\Phi$, we obtain that $\Phi(G \setminus B', G)$ holds, contradicting $B'$ being a blocking-set. Hence, $C \cap B' \neq \emptyset$, so $B'$ is a hitting set of $\mathcal{C}$. But since $|B'| \leq |B|$, this contradicts the minimality of $B$ as an MHS. Therefore, $B$ is cardinally minimal among blocking-sets.

$\square$

## B.8 PROOF OF PROPOSITION 8

**Proposition 8.** *Given a model $f_G$, and a monotonic predicate $\Phi$, Algorithm 2 computes a subset $C$ whose size is a* lower bound *to the cardinally minimal circuit for which $\Phi(C, G)$ is true. For a large enough $t_{max}$ value, the algorithm converges* exactly *to the cardinally minimal circuit.*

*Proof.* We begin with the proof for the part on the lower bound to cardinally minimal circuit size. Let $C$ be the output of Algorithm 2 for some $t_{\max}$. By definition, $C$ is the MHS of the set of blocking-sets accumulated by the algorithm, denoted $\mathcal{B}_{t_{\max}}$.

Assume towards contradiction that $|C|$ is not a lower bound for the size of a cardinally minimal circuit. This would imply the existence of a faithful circuit $C'$ with $\Phi(C', G)$ and $|C'| \leq |C|$. From the minimality of $C$ as a hitting set, it follows that $C'$ is not a hitting set. Hence, there exists some $B \in \mathcal{B}_{t_{\max}}$ such that $C' \cap B = \emptyset$. This implies $C' \subseteq G \setminus B$, and by monotonicity of $\Phi$ we obtain $\Phi(G \setminus B, G)$, contradicting $B$ being a blocking-set.

We now continue to the second part of the proof regarding the convergence to cardinally minimal for large enough $t_{\max}$. For $t_{\max} = |G|$, the algorithm iterates over all possible blocking-sets. Hence, the resulting output $C$ is the MHS of all circuit blocking-sets, and by Proposition 7 we conclude that $C$ is a cardinally minimal circuit.

$\square$

## C  Minimality Guarantees: Algorithms and Illustrations

### C.1  Greedy Exhaustive Iterative Circuit Discovery

We present here an algorithm for exhaustively and repeatedly iterating over the circuit components, removing any component whose deletion does not violate faithfulness, until no further removals are possible (Algorithm 3).

---

**Algorithm 3** Greedy Exhaustive Circuit Discovery

---

 1: **Input** Model $f_G$, circuit faithfulness predicate $\Phi$
 2: $C \leftarrow G$ under some given element ordering (e.g., reverse topological sort)
 3: $changed \leftarrow$ **true**
 4: **while** $changed$ **do**
 5:     $changed \leftarrow$ **false**
 6:     **for all** $i \in C$ **do**
 7:         **if** $\Phi(C \setminus \{i\}, G)$ **then**
 8:             $C \leftarrow C \setminus \{i\}$
 9:             $changed \leftarrow$ **true**
10:         **end if**
11:     **end for**
12: **end while**
13: **return** $C$

---

### C.2  Greedy Circuit Discovery Binary Search for Quasi-Minimal circuits

We formalize the binary search procedure introduced in (Adolfi et al., 2025) in Algorithm 4.

---

**Algorithm 4** Greedy Circuit Discovery Binary Search

---

 1: **Input:** Model $f_G$, circuit faithfulness predicate $\Phi$ **with** $\Phi(G, G) \ \wedge \ \neg\Phi(\emptyset, G)$
 2: $C \leftarrow G$, low $\leftarrow 0$, high $\leftarrow |G|$
 3: **while** high $-$ low $> 1$ **do**
 4:     mid $\leftarrow \lfloor (\text{low} + \text{high})/2 \rfloor$
 5:     $C_{\text{mid}} \leftarrow G \setminus G[1 : \text{mid}]$
 6:     **if** $\Phi(C_{\text{mid}}, G)$ **then**
 7:         low $\leftarrow$ mid; $C \leftarrow C_{\text{mid}}$
 8:     **else**
 9:         high $\leftarrow$ mid
10:     **end if**
11: **end while**
12: **return** $C$

---

### C.3  Toy Example: Minimality Notions

To illustrate the distinctions between the four minimality notions introduced in Definitions 3,4,5,6 (Section 4), we construct a simple Boolean toy network.

For simplicity, we illustrate the different minimality notions using a Boolean circuit with XOR gates. While this abstraction makes the example easier to follow, it is without loss of generality, since Boolean gates can be equivalently expressed with ReLU activations.

(Specifically, the XOR gate satisfies

$$x_1 \oplus x_2 \ = \ \text{ReLU}(x_1 - x_2) + \text{ReLU}(x_2 - x_1),$$

as for $x_1, x_2 \in \{0, 1\}$ both terms vanish when $x_1 = x_2$, and exactly one equals 1 when $x_1 \neq x_2$).

We emphasize that this encoding is not part of the computation graph, which can be defined independently. Accordingly, our toy boolean circuit (Fig. 6) can be viewed as a small feed-forward ReLU

network. Despite its small size, this network cleanly separates the notions of *cardinal*, *subset*, *local*, and *quasi*-minimal circuits.

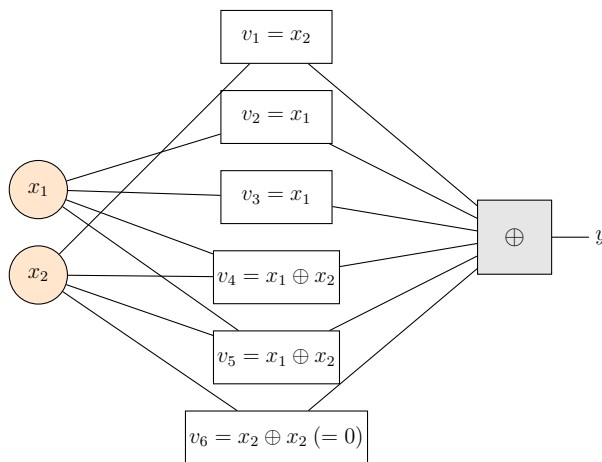

Figure 6: Boolean toy network with XOR aggregation: $y = v_1 \oplus v_2 \oplus v_3 \oplus v_4 \oplus v_5$. With $v_1 = x_2$, $v_2 = v_3 = x_1$, $v_4 = v_5 = x_1 \oplus x_2$, and $v_6 = x_2 \oplus x_2 = 0$ the full model computes $f_G(x_1, x_2) = x_2$.

**Explanation.** For clarity and simplicity, we define a Boolean network $G$ which takes inputs in $(x_1, x_2) \in \{0,1\}^2$. The network $G$ is composed of six components, whose vertex set is denoted $V_G = \{v_1, v_2, v_3, v_4, v_5, v_6\}$, aggregated by XOR (Fig. 6):

$$y = v_1 \oplus v_2 \oplus v_3 \oplus v_4 \oplus v_5 \oplus v_6, \quad \text{with} \quad v_1 = x_2, \ v_2 = v_3 = x_1, \ v_4 = v_5 = x_1 \oplus x_2, v_6 = x_2 \oplus x_2$$

The full network computes $f_G(x_1, x_2) = x_2$, since

$$f_G(x_1, x_2) = x_2 \oplus x_1 \oplus x_1 \oplus (x_1 \oplus x_2) \oplus (x_1 \oplus x_2) \oplus (x_2 \oplus x_2)$$
$$= x_2 \oplus \underbrace{(x_1 \oplus x_1)}_{=0} \oplus \underbrace{[(x_1 \oplus x_2) \oplus (x_1 \oplus x_2)]}_{=0} \oplus \underbrace{(x_2 \oplus x_2)}_{=0} = x_2.$$

We pick the faithfulness predicate to be

$$\Phi(C, G) := f_C(x_1, x_2) = f_G(x_1, x_2) = x_2 \quad \forall(x_1, x_2),$$

i.e., $C$ is faithful if it computes the same output as $G$ on all inputs.

This single construction cleanly separates the four minimality notions:

For each circuit $C$, we verify that it satisfies $f_C(x_1, x_2) = x_2$ and state why it meets (or fails) the corresponding minimality condition.

- **Cardinal-minimal:** $C_{\text{card}} = \{v_1\}$. It computes $f_{C_{\text{card}}} = v_1 = x_2 = f_G$. No circuit with fewer components can be faithful.
- **Subset-minimal:** $C_{\text{sub}} = \{v_2, v_4\}$. The computation is
  $$v_2 \oplus v_4 = x_1 \oplus (x_1 \oplus x_2) = x_2.$$
  Removing any component breaks correctness: $\{v_2\} = x_1 \neq x_2$ and $\{v_4\} = x_1 \oplus x_2 \neq x_2$.
- **Local-minimal:** $C_{\text{loc}} = \{v_1, v_2, v_3\}$. It computes
  $$v_1 \oplus v_2 \oplus v_3 = x_2 \oplus x_1 \oplus x_1 = x_2.$$
  Removing any single component breaks correctness: $\{v_1, v_2\} = x_2 \oplus x_1 \neq x_2$, $\{v_1, v_3\} = x_2 \oplus x_1 \neq x_2$, $\{v_2, v_3\} = x_1 \oplus x_1 = 0 \neq x_2$. However, removing two components may still leave a correct singleton (e.g. $\{v_1\}$), so it is not subset-minimal.
- **Quasi-minimal:** $C_{\text{quasi}} = \{v_1, v_2, v_3, v_4, v_5\}$. It computes
  $$v_1 \oplus v_2 \oplus v_3 \oplus v_4 \oplus v_5 = x_2.$$
  The circuit contains a single essential component ($v_1$), while the remaining ones can be removed in various combinations without changing the output (e.g., $v_2 \oplus v_3 = 0$ and $v_4 \oplus v_5 = 0$). Hence it is faithful but not minimal under any stricter notion.

# D  BENCHMARKS, MODELS AND ARCHITECTURAL SPECIFICATIONS

We evaluate our methods on four standard benchmarks in neural network verification: three classification benchmarks (CIFAR-10 (Krizhevsky & Hinton, 2009), GTSRB (Stallkamp et al., 2011), and MNIST (Lecun et al., 1998)) and one regression benchmark, TaxiNet (Julian et al., 2020).

For each benchmark, we perform circuit discovery at the natural level of granularity for the model: convolutional filters (or *channels*) in CNNs and neurons in the fully connected network. The number of components at each granularity is summarized in Table 3.

Table 3: Granularity and number of components considered for circuit discovery across the benchmark models.

| Dataset | Model | Examined Granularity | # Components |
|---------|-------|---------------------|--------------|
| MNIST | FC | Neurons | 31 |
| GTSRB | CNN | Filters | 48 |
| CIFAR-10 | ResNet | Filters | 72 |
| TaxiNet | CNN | Filters | 8 |

**Data Selection.**    For the input and patching robustness experiments (Appendices E, F), we constructed at least 100 batches per benchmark, sampled from the **test** set using only correctly predicted inputs (or low-error inputs in the regression case). In classification tasks, each batch contained $k = 3$ samples from a single class, evenly distributed across classes. In the regression task, batches of $k = 3$ were drawn from inputs with absolute error below 0.2, excluding large deviations relative to the model's performance. Specifically, we sampled 100 batches for **CIFAR-10** and **MNIST** (10 per class), 129 batches for **GTSRB** (3 per class across 43 classes), and 100 batches for the regression benchmark **TaxiNet**.

For the minimality guarantees experiment (Subsection 5.3, Appendix G), we used 50 singleton batches ($k = 1$), obtained by selecting one sample from each MNIST batch above, thereby preserving the even class distribution.

## D.1  CIFAR-10

For the CIFAR-10 benchmark Krizhevsky & Hinton (2009), we use the ResNet2b model, originating from the VNN-COMP neural network verification competition (Bak et al., 2021). This residual network consists of an initial convolutional layer, two residual blocks, and a dense classification head producing 10 output classes. In total, it comprises 72 filters (also referred to as *channels*).

| Layer / Block | Output Dim. | Details |
|---|---|---|
| Input | $32 \times 32 \times 3$ | CIFAR-10 image |
| Conv1 | $16 \times 16 \times 8$ | $3 \times 3$, stride 2 |
| *Residual Block 1* | | |
|    Conv1 | $8 \times 8 \times 16$ | $3 \times 3$, stride 2 |
|    ReLU | $8 \times 8 \times 16$ | non-linearity |
|    Conv2 | $8 \times 8 \times 16$ | $3 \times 3$, stride 1 |
|    Skip connection | $8 \times 8 \times 16$ | identity/projection |
|    Output | $8 \times 8 \times 16$ | addition + ReLU |
| *Residual Block 2* | | |
|    Conv1 | $8 \times 8 \times 16$ | $3 \times 3$, stride 1 |
|    ReLU | $8 \times 8 \times 16$ | non-linearity |
|    Conv2 | $8 \times 8 \times 16$ | $3 \times 3$, stride 1 |
|    Skip connection | $8 \times 8 \times 16$ | identity |
|    Output | $8 \times 8 \times 16$ | addition + ReLU |
| Flatten | $1 \times 2048$ | – |
| Linear1 | $2048 \rightarrow 100$ | ReLU |
| Linear2 | $100 \rightarrow 10$ | Output logits |

Table 4: Full architecture of the ResNet2b model used in Bak et al. (2021) for the CIFAR-10 benchmark

## D.2 GTSRB

The German Traffic Sign Recognition Benchmark (GTSRB) Stallkamp et al. (2011) is a large-scale image classification dataset containing more than 50,000 images of traffic signs across 43 classes, captured under varying lighting and weather conditions.

We adopt the GTSRB-CNN model used in recent explainability studies (Bassan et al., 2025a). This architecture is a convolutional network with two convolutional layers using ReLU activations and average pooling, followed by two fully connected layers. It outputs logits over 43 traffic sign classes.

In total, the GTSRB-CNN comprises 48 filters across its two convolutional layers (16 + 32).

| Layer | Output Dim. | Details |
|---|---|---|
| Input | $32 \times 32 \times 3$ | GTSRB image |
| Conv1 | $32 \times 32 \times 16$ | $3 \times 3$, padding 1, ReLU |
| AvgPool1 | $16 \times 16 \times 16$ | $2 \times 2$ |
| Conv2 | $16 \times 16 \times 32$ | $3 \times 3$, padding 1, ReLU |
| AvgPool2 | $8 \times 8 \times 32$ | $2 \times 2$ |
| Flatten | $1 \times 2048$ | – |
| FC1 | $2048 \rightarrow 128$ | ReLU |
| FC2 | $128 \rightarrow 43$ | Output logits |

Table 5: Architecture of the GTSRB-CNN model used in (Bassan et al., 2025a).

## D.3 MNIST

We use a simple, classic fully connected feedforward network for MNIST classification, which we trained given the simplicity of the task. The model achieves 95.20% accuracy on the test set. It consists of two hidden layers with ReLU activations of sizes 13 and 11, followed by a linear output layer, comprising **31** non-input neurons in total and 10,479 trainable parameters.

| Layer | Dimensions | Activation |
|---|---|---|
| Input | $28 \times 28 = 784$ | – |
| Fully Connected (fc1) | $784 \rightarrow 13$ | ReLU |
| Fully Connected (fc2) | $13 \rightarrow 11$ | ReLU |
| Fully Connected (fc3) | $11 \rightarrow 10$ | – |

Table 6: Architecture of the fully connected MNIST network. The model has 31 hidden neurons in total, which we treat as the granularity for circuit discovery.

### D.4 TAXINET

The TaxiNet dataset (Julian et al., 2020) was developed by NASA for vision-based aircraft taxiing, and consists of synthetic runway images paired with continuous control targets. Unlike the classification benchmarks, TaxiNet is a *regression* task: the model predicts real-valued outputs corresponding to flight control variables.

For our experiments, we adopt the TaxiNet CNN regression model introduced in the VeriX framework (Wu et al., 2023a) and subsequently used in other explainability studies (Bassan et al., 2025a). This convolutional network, comprising 8 filters, achieves a mean squared error (MSE) of 0.848244, and a root mean squared error (RMSE) of 0.921.

| Layer | Output Dim. | Details |
|---|---|---|
| Input | $27 \times 54 \times 1$ | TaxiNet image |
| Conv1 | $27 \times 54 \times 4$ | $3 \times 3$, padding 1, ReLU |
| Conv2 | $27 \times 54 \times 4$ | $3 \times 3$, padding 1, ReLU |
| Flatten | $1 \times 5832$ | – |
| FC1 | $5832 \rightarrow 20$ | ReLU |
| FC2 | $20 \rightarrow 10$ | ReLU |
| FC3 | $10 \rightarrow 1$ | Regression output |

Table 7: Architecture of the CNN regression model used for TaxiNet, following the VeriX framework (Wu et al., 2023a).

## EXPERIMENTAL DETAILS

## E    INPUT ROBUSTNESS CERTIFICATION

In this experiment, we evaluate the robustness of discovered circuits over a continuous input neighborhood $\mathcal{B}_\epsilon^p(\mathbf{x})$, as established in Section 3.1. We compare two variants of the iterative circuit discovery procedure in Algorithm 1, which differ in their elimination criterion:

1. **Sampling-based Circuit Discovery**: directly evaluates the metric on the input batch at each step.

2. **Provably Input-Robust Circuit Discovery**: certifies that the metric holds across the entire input neighborhood (Def. 1).

The procedure traverses network components sequentially, deciding at each step whether to retain or remove a component. As noted in Conmy et al. (2023), the traversal order influences the resulting circuit. Following their approach, we proceed from later fully-connected or convolutional layers toward earlier ones, ordering neurons or filters within each layer lexicographically. For consistency, we fix the patching scheme for all non-circuit components to *zero-patching*.

### E.1 METHODOLOGY

#### E.1.1 SIAMESE NETWORK FOR VERIFICATION

To integrate circuit discovery with formal verification, we construct a *Siamese Network*, which pairs the full model with a candidate circuit, and outputs the **concatenation** of the two networks' logits.

This Siamese formulation provides the interface for the neural network verification used in two settings:

- **Provably Input-Robust Circuit Discovery:** certify at each elimination step that the candidate circuit satisfies the metric across the continuous input neighborhood, ensuring robustness is preserved.
- **Evaluation:** verify after discovery (via either sampling- or provable-based methods) that the resulting circuit is robust over the same neighborhood.

**Output Metric.** For consistency, all of our experiments employ the same output metric for both the sampling-based and provably input-robust discovery methods. We measure the difference in logits, requiring this difference to remain within a tolerance $\delta$.

In classification tasks over an input $z$, we focus on the logit of the gold-class, indexed by $k$, and require that the predictions of the circuit $C$ and the full model $G$ differ by at most $\delta$:

$$|f_G(z)[k] - f_C(z)[k]| \leq \delta,$$

where the absolute value denotes the $\ell_p$-norm on the one-dimensional vector corresponding to the $k$-th entry. In the sampling-based method, it is evaluated on the sampled batch, whereas in the provable (siamese) setting, this criterion is certified over the concatenated logits of the siamese encoding. For instance, in a 10-class classification task, the verification constraint on the siamese network's output is:

$$|\, \text{logits}_{[:10]}[k] - \text{logits}_{[10:]}[k]\, | \leq \delta.$$

where the first 10 entries correspond to the logits of the full model $G$ and the second to those of the circuit $C$.

In regression settings (e.g., TaxiNet), the same principle applies to the full output (a scalar-valued prediction), measuring the absolute difference between the model and the circuit. In both cases, this metric directly instantiates the norm metric used in the robustness definitions (Definitions 1, 2).

**Input Neighborhoods.** In our setup, the neighborhood $\mathcal{B}^\infty_{\epsilon_p}(\mathbf{x})$ is defined in the input space with respect to the $L^\infty$ norm. For fully connected models (e.g., MNIST), inputs are flattened into vectors and the perturbation ball is defined over this representation. For convolutional models (e.g., CIFAR-10, GTSRB, TaxiNet), inputs are multi-channel tensors, and the neighborhood is applied independently to each channel and spatial location.

#### E.1.2 VERIFICATION AND EXPERIMENTAL SETUP (INPUT ROBUSTNESS)

Since sound-and-complete verification of piecewise linear activation networks against linear properties is NP-hard Katz et al. (2017), Some queries may not complete within the allotted time; in such cases, the outcome is reported as *unknown*. In practice, with the $\alpha, \beta$-CROWN verifier, we limit each query to **45** seconds of Branch-and-bound time.

For fairness, we report robustness statistics only on batches where the robustness check of the sampling-based method was determined (robust or non-robust, excluding timeouts) In the main paper results, the rate of timed-out instances was 1% on MNIST, 1.6% on GTSRB, 1% on CIFAR-10, and 5% on TaxiNet. Comparable rates were observed in the neighborhood size variations E.2.1 studies (on average, 0.5% for MNIST, 3% for CIFAR-10, and 2.7% for TaxiNet), and in the tolerance level variations $\delta$ E.2.2 (on average 8.6% for TaxiNet, 3.6% for MNIST).

Experiments on MNIST, GTSRB, and CIFAR-10 were conducted on a unified hardware setup with a 48 GB NVIDIA L40S GPU paired with a 2-core, 16 GB CPU. For the TaxiNet model, we used only a 2-core, 36 GB CPU machine.

### E.2 ROBUSTNESS EVALUATION AND PARAMETER VARIATIONS

We evaluate the robustness of circuits discovered by both methods over the neighborhood $\mathcal{B}_{\epsilon_p}^{\infty}(\mathbf{x})$, using formal verification via the *Siamese Network* as described above. The parameter $\epsilon_p$ controls the neighborhood size: if too small, the resulting circuits are trivial; if too large, the perturbations become unrealistic and off-distribution. We choose $\epsilon_p$ values within ranges commonly used in prior verification work or empirically selected to balance circuit size and robustness. In addition, we vary both parameters, $\epsilon_p$ and $\delta$, to analyze their effect.

#### E.2.1 VARIATION OF INPUT NEIGHBORHOOD SIZE $\epsilon_p$

We fix the tolerance $\delta$ and vary the input neighborhood size $\epsilon_p$. For CIFAR-10 and MNIST, we use $\delta = 2.0$; for GTSRB, $\delta = 5.0$. For the TaxiNet regression model, we set $\delta = 0.92$ (the model's root mean squared error, RMSE), reflecting its typical prediction scale (larger deviations would let the circuit drift more than the full model from the ground truth). Results are reported in Table 8. Rows highlighted in gray correspond to the results selected in the main paper.

| Dataset | $\epsilon_p$ | Sampling-based Circuit Discovery | | | Provably Input-Robust Circuit Discovery | | |
|---|---|---|---|---|---|---|---|
| | | Time (s) | Size ($|C|$) | Robustness (%) | Time (s) | Size ($|C|$) | Robustness (%) |
| MNIST ($\delta$=2.0) | | | | | | | |
| | 0.005 | 0.015 ±0.003 | 12.57 ±2.29 | **46.0** ±5.0 | 677.36 ±183.65 | 14.51 ±2.41 | **100.0** ±0.0 |
| | 0.009 | 0.016 ±0.013 | 12.56 ±2.30 | **25.3** ±4.4 | 663.60 ±181.13 | 15.76 ±2.25 | **100.0** ±0.0 |
| | 0.010 | 0.309 ±0.889 | 12.56 ±2.30 | **19.2** ±4.0 | 611.93 ±97.14 | 15.84 ±2.33 | **100.0** ±0.0 |
| | 0.050 | 0.027 ±0.065 | 12.57 ±2.29 | **0.0** ±0.0 | 1700.05 ±562.36 | 28.75 ±6.67 | **100.0** ±0.0 |
| TaxiNet ($\delta$=0.92) | | | | | | | |
| | 0.001 | 0.040 ±0.146 | 5.76 ±0.77 | **62.2** ±4.9 | 201.81 ±34.16 | 6.07 ±0.71 | **100.0** ±0.0 |
| | 0.005 | 0.010 ±0.002 | 5.77 ±0.80 | **9.5** ±3.0 | 180.00 ±40.39 | 6.82 ±0.46 | **100.0** ±0.0 |
| | 0.010 | 0.010 ±0.002 | 5.78 ±0.79 | **2.0** ±1.4 | 271.03 ±54.23 | 7.91 ±0.32 | **100.0** ±0.0 |
| CIFAR-10 ($\delta$=2.0) | | | | | | | |
| | 0.007 | 0.035 ±0.001 | 16.70 ±9.48 | **73.7** ±4.4 | 2104.13 ±118.95 | 17.96 ±9.90 | **100.0** ±0.0 |
| | 0.012 | 0.116 ±0.367 | 16.91 ±9.12 | **58.1** ±5.1 | 2226.34 ±103.71 | 18.88 ±9.21 | **100.0** ±0.0 |
| | 0.015 | 0.228 ±0.517 | 16.47 ±9.08 | **46.5** ±5.0 | 2970.85 ±874.23 | 19.18 ±10.16 | **100.0** ±0.0 |
| GTSRB ($\delta$=5.0) | | | | | | | |
| | 0.001 | 0.111 ±0.329 | 28.91 ±4.69 | **27.6** ±4.0 | 991.08 ±162.91 | 29.59 ±4.45 | **100.0** ±0.0 |

Table 8: Effect of varying the input neighborhood size $\epsilon_p$ under a fixed tolerance $\delta$. Reported values are means with standard deviations. For robustness (a binary variable), we report the standard error (SE). Bold values indicate robustness percentages. Rows highlighted in gray correspond to the results selected in the main paper.

#### E.2.2 VARIATION OF TOLERANCE LEVEL $\delta$

We vary the tolerance $\delta$ while fixing the input neighborhood size $\epsilon_p$ to the dataset-specific values used in the main paper (MNIST: $\epsilon_p$=0.01, TaxiNet: $\epsilon_p$=0.005). Results are reported in Table 9. Rows corresponding to the main paper results are highlighted in gray.

| Dataset | $\delta$ | Sampling-based Circuit Discovery | | | Provably Input-Robust Circuit Discovery | | |
|---|---|---|---|---|---|---|---|
| | | Time (s) | Size ($|C|$) | Robustness (%) | Time (s) | Size ($|C|$) | Robustness (%) |
| MNIST ($\epsilon_p$=0.01) | | | | | | | |
| | 0.50 | 0.083 ± 0.329 | 19.72 ± 1.50 | **15.6** ± 3.8 | 460.14 ± 36.59 | 22.02 ± 2.52 | **100.0** ± 0.0 |
| | 2.00 | 0.309 ± 0.889 | 12.56 ± 2.29 | **19.2** ± 4.0 | 611.93 ± 97.14 | 15.84 ± 2.33 | **100.0** ± 0.0 |
| | 3.00 | 0.013 ± 0.001 | 9.44 ± 1.92 | **34.0** ± 4.7 | 577.41 ± 35.84 | 12.27 ± 2.52 | **100.0** ± 0.0 |
| TaxiNet ($\epsilon_p$=0.005) | | | | | | | |
| | 0.50 | 0.038 ± 0.115 | 6.84 ± 0.86 | **32.1** ± 5.2 | 93.08 ± 22.75 | 7.75 ± 0.46 | **100.0** ± 0.0 |
| | 0.70 | 0.008 ± 0.001 | 6.15 ± 0.81 | **14.8** ± 3.8 | 114.51 ± 23.95 | 7.31 ± 0.53 | **100.0** ± 0.0 |
| | 0.92 | 0.010 ± 0.002 | 5.77 ± 0.80 | **9.5** ± 3.0 | 180.00 ± 40.39 | 6.82 ± 0.46 | **100.0** ± 0.0 |
| | 1.00 | 0.009 ± 0.002 | 5.57 ± 0.82 | **9.5** ± 3.0 | 142.62 ± 24.77 | 6.66 ± 0.52 | **100.0** ± 0.0 |
| | 1.20 | 0.009 ± 0.001 | 5.43 ± 0.96 | **6.1** ± 2.4 | 155.03 ± 26.34 | 6.32 ± 0.59 | **100.0** ± 0.0 |

Table 9: Variation on tolerance level $\delta$, with input neighborhood size $\epsilon$ fixed to the dataset-specific values used in the main experiments. Reported values are means with standard deviations. For robustness (a binary variable), we report the standard error (SE). Rows highlighted in gray correspond to the results selected in the main paper.

### E.2.3 EVALUATION UNDER ALTERNATIVE OUTPUT METRICS

To further assess the generality of our framework, we repeat the robustness evaluation using alternative output metrics beyond the default *logit-difference* criterion. Specifically, we consider two additional formulations:

- **Consistent winner class:** Given some target class $t \in [d]$, enforces that the winner class remains consistent over a specified region. This metric directly targets preservation of the predicted class across the input domain and is widely used in robustness verification studies. The criterion then enforces that:

$$\forall \mathbf{z} \in \mathcal{Z}, \quad \operatorname*{argmax}_{j}(f_G(\mathbf{z}))_{(j)} = \operatorname*{argmax}_{j}(f_C(\mathbf{x} \mid \overline{C} = \alpha)(\mathbf{z}))_{(j)} = t.$$

  To allow greater flexibility, we relax the requirement by permitting the predicted class to remain consistent under any change that stays within a tolerance $\delta \in \mathbb{R}^+$ above the runner-up class. When $\delta = 0$, this reduces back to the original definition. To make this threshold more meanignful and interpretable, we set $\delta$ as a configurable fraction $\alpha \in (0, 1]$ of the model's original winner–runner gap on the unperturbed input. In our experiments, for simplicity, we enforce the consistency condition only between the winner and runner-up classes.

- **Abs-Max:** Bounds the maximum absolute deviation across all output dimensions by a specified threshold. This criterion does not guarantee class invariance but constrains the overall output drift. Formally, we require:

$$\forall \mathbf{z} \in \mathcal{Z}, \quad \|f_G(\mathbf{z}) - f_C(\mathbf{x} \mid \overline{C} = \alpha)\|_\infty \leq \delta,$$

  ensuring that no individual logit differs by more than $\delta$.

We evaluate both metrics, using the same discovery configurations and $\epsilon_p$ as in the main input-robustness experiments. For the logits-difference metric, we used the same $\delta$ as in our main experiment (Table 1). In the winner-runner setting, we set $\alpha = 0.5$ (preserving half the original margin), and for the abs-max criterion we used $\delta = 4.0$. Results are reported in Table 10, which compares the *sampling-based* and *provable* discovery methods under each metric.

| Dataset | Metric | Sampling-based Circuit Discovery | | | Provably Input-Robust Circuit Discovery | | |
|---|---|---|---|---|---|---|---|
| | | Time (s) | Size ($|C|$) | Robustness (%) | Time (s) | Size ($|C|$) | Robustness (%) |
| MNIST | Logit-diff | 0.31 ±0.89 | 12.56 ±2.30 | **19.2** ±4.0 | 611.93 ±97.14 | 15.84 ±2.33 | **100.0** ±0.0 |
| | Winner–Runner | 0.13 ± 0.60 | 5.18 ± 1.05 | **73.0** ±4.4 | 638.57 ± 20.94 | 12.04 ± 5.79 | **100.0** ±0.0 |
| | Abs-Max | 0.014 ± 0.012 | 25.55 ± 2.90 | **6.0** ±2.4 | 362.61 ± 55.73 | 28.11 ± 2.37 | **100.0** ±0.0 |

Table 10: Comparison of circuit discovery methods under alternative output metrics. Reported values are means with standard deviations. For robustness (a binary variable), we report the standard error (SE). All methods use the same configurations as in the main experiments.

Across all metrics, the same overall trend is observed: the provable method consistently approaches 100% robustness while maintaining circuit sizes comparable to those of the sampling-based baseline, which attains substantially lower robustness. This consistency across different metrics suggests that the robustness of the provable approach is not tied to a particular output metric, but reflects a stable characteristic of the method.

### E.2.4 COVERAGE ANALYSIS OF PROVABLY-ROBUST VS. SAMPLING-BASED CIRCUITS

To better understand the relationship between the circuits identified by our provably-robust procedure and those produced by the sampling-based method, we conduct an explicit coverage analysis over several robustness radii $\epsilon \in \{0.005, 0.007, 0.01, 0.02, 0.03, 0.04, 0.05\}$ on the MNIST benchmark. All other settings, including tolerance and metric definitions, follow those used in the main experiment (as discussed in section E.1). We conduct these experiments on a 2-core CPU machine with 16 GB of RAM.

For each perturbation radius $\epsilon_p$, we examine the provably input-robust circuit $C_p$ derived for that radius and the sampling-based circuit $C_s$, each obtained over 100 different inputs (as in our experimental setup), resulting in 100 circuit pairs for every $\epsilon_p$.

For these two circuits, we compute: (i) the size of the intersection, $|C_p \cap C_s|$, (ii) the components unique to the provable circuit (*provable-only*), $|C_p \setminus C_s|$, and (iii) the components unique to the sampling-based circuit (*sampling-only*), $|C_s \setminus C_p|$.

We average these quantities over the 100 circuit pairs and report their means and standard deviations. To summarize the overall similarity/discrepancy between $C_p$ and $C_s$ across $\epsilon_p$, we additionally compute standard set-similarity measures: Intersection over Union (IoU), Dice coefficient, and two asymmetric coverage metrics (provable-over-sampling and sampling-over-provable). These aggregate trends are visualized in Fig. 7. To further highlight the non-overlapping components, Table 11 reports their counts and their percentages relative to the full network size.

Because the sampling-based method does not enforce a robustness condition, its circuit size remains constant across $\epsilon_p$, while the provable-based circuits naturally expand as $\epsilon_p$ increases in order to guarantee certified robustness. We indeed view that as the required robustness grows, the provably-robust circuits include additional components essential for certification.

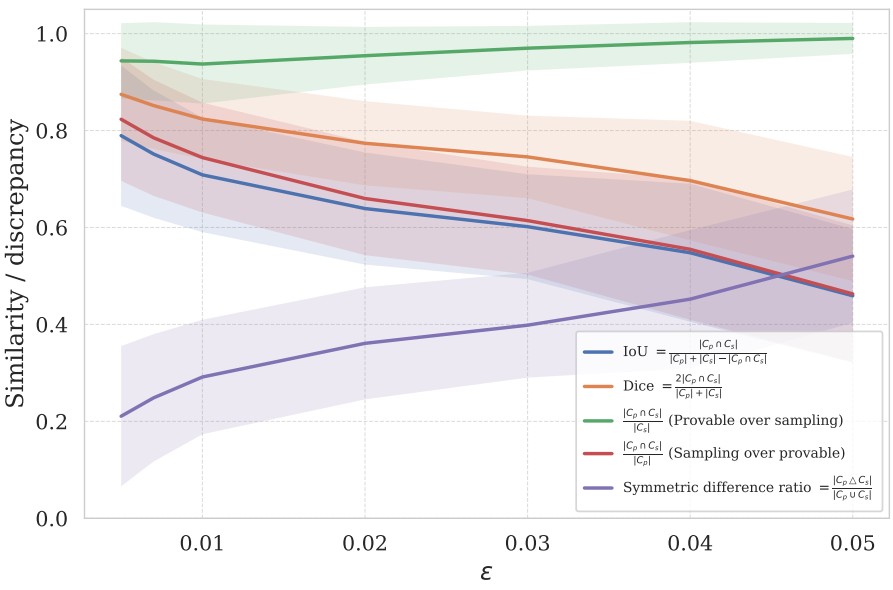

Figure 7: Comparison of similarity and coverage metrics between provably-robust and sampling-based circuits. We report symmetric measures (IoU, Dice), a symmetric difference ratio, and asymmetric coverage ratios (*provable over sampling*, *sampling over provable*) to illustrate both overlap and directional differences.

As shown in Fig. 7, the provably robust circuits consistently recover the vast majority of units identified by the sampling-based method across all $\epsilon_p$ values, with especially high agreement for small perturbation radii (e.g., IoU and Dice $\approx 0.9$ at $\epsilon_p = 0.005$). As $\epsilon_p$ increases, the overlap between the two circuits gradually decreases (IoU drops toward 0.5), indicating that the sampling-based circuits capture a smaller fraction of the provable-based ones under larger perturbations.

| Dataset | $\epsilon_p$ | $|C_p \setminus C_s|$ | % of full net | $|C_s \setminus C_p|$ | % of full net |
|---------|-----|------|--------|------|-------|
| | 0.005 | 2.64 | 7.76% | 0.70 | 2.06% |
| | 0.007 | 3.32 | 9.76% | 0.74 | 2.18% |
| | 0.010 | 4.10 | 12.06% | 0.81 | 2.38% |
| MNIST | 0.020 | 6.24 | 18.35% | 0.58 | 1.71% |
| | 0.030 | 7.73 | 22.74% | 0.39 | 1.15% |
| | 0.040 | 10.94 | 32.18% | 0.24 | 0.71% |
| | 0.050 | 16.11 | 47.38% | 0.13 | 0.38% |

Table 11: Set differences between the provably-robust circuit $C_p$ and the sampling-based circuit $C_s$. For each $\epsilon_p$, we report (i) the number of units appearing only in the provably-robust circuit $|C_p \setminus C_s|$, (ii) the number appearing only in the sampling-based circuit $|C_s \setminus C_p|$, and (iii) the corresponding percentages relative to the full network size for that dataset.

This reflects the fact that the provably-robust circuits expand to satisfy stronger robustness requirements. This trend is also evident in Table 11: the difference $C_p \setminus C_s$ grows steadily with $\epsilon_p$, while $C_s \setminus C_p$ remains small across all settings, and decreases further for larger perturbation radii - indicating that the sampling-based method contributes few components that are not required by the provable, certified solution.

### E.2.5 RUNTIME TRADE-OFF ACROSS INPUT NEIGHBORHOOD SIZES

We aim to further analyze the runtime trade-off of the provably-robust method. For each input-robustness radius $\epsilon_p$, we run the method to obtain a corresponding provably robust circuit and report the mean circuit-size-over-time curves (with standard deviation shown as shaded regions) across these circuits for different input neighborhoods induced by increasing $\epsilon_p$. We perform this analysis on the MNIST benchmark, using the same perturbation radii and experimental settings as in the coverage analysis in Appendix E.2.4.

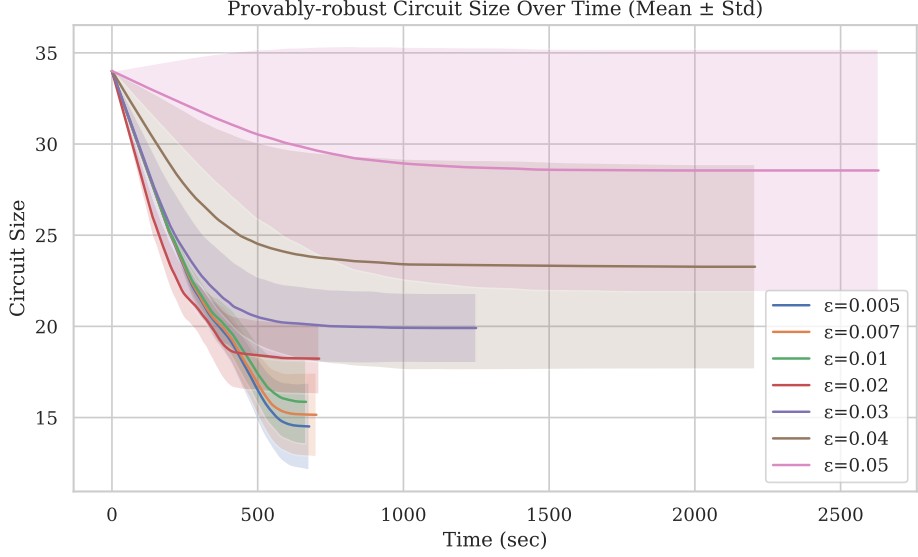

Figure 8: Provable-robust circuit size over time on MNIST for different neighborhood radii $\epsilon_p$. Shaded regions denote the standard deviation.

In all cases, the curves start at the full network size at time 0 and then decrease monotonically as components are pruned, until the procedure terminates. Across the smaller and closely spaced $\epsilon_p$ values, the curves exhibit a very similar trajectory: an initial almost-linear decrease during the first $\sim 300$ seconds, followed by a stabilization phase around $\sim 500$ seconds. The standard deviation

bands for these curves are also comparable. As expected, larger neighborhoods lead to larger final circuit sizes.

For substantially larger neighborhoods (e.g., $\epsilon_p = 0.03\text{-}0.05$), the behavior changes: the decrease is slower, stabilization occurs later, and the variability (standard deviation) is considerably higher. Moreover, for these larger and more widely spaced $\epsilon_p$ values, we observe a clear increase in overall runtime (as indicated by where the curves terminate), reflecting the added complexity of discovering robust certified circuits under broader perturbation regions.

### E.2.6 QUALITATIVE OBSERVATIONS OF THE DISCOVERED CIRCUITS

While our method is centered on formal guarantees, and our evaluation therefore focuses on robustness and minimality, we also include a brief exploratory look at the circuits discovered by our provably robust procedure and by the sampling-based baseline. This examination is qualitative in nature and is intended only to provide an informal visual sense of how the two circuits behave.

For this analysis, we consider several channel-level GTSRB circuits produced in the input-robustness experiments. Recall that for each batch (composed of samples from the same class) we executed both our provably robust discovery (under a given $\epsilon = 0.001$ neighborhood) and the sampling-based discovery, producing two circuits. In the examples below, we select pairs of circuits with comparable sizes, where the sampling-based circuit is empirically non-robust while the provably robust circuit is certified robust. We then analyze their behavior on a representative clean input from the batch and on its corresponding adversarial example (an $\epsilon = 0.001$-bounded adversarial perturbation).

To obtain a coarse semantic signal, we apply Grad-CAM (Selvaraju et al., 2017) to the last convolutional layer of (i) the full model, (ii) the provably robust circuit, and (iii) the sampling-based circuit. Grad-CAM produces a class-specific importance map by weighting spatial activations according to the globally averaged gradients of the target class logit. Formally, for class $c$,

$$\alpha_k^{(c)} := \frac{1}{HW} \sum_{i,j} \frac{\partial y_c}{\partial A_{ij}^k}, \qquad \mathrm{CAM}_c(i,j) := \mathrm{ReLU}\left( \sum_k \alpha_k^{(c)} A_{ij}^k \right).$$

Here, $y_c$ denotes the logit of the target class $c$ (the true label in our case), $A^k$ is the $k$-th activation map (i.e., the output of filter $k$) of spatial size $H \times W$, and $\alpha_k^{(c)}$ is the Grad-CAM weight obtained by spatially averaging the gradients $\partial y_c / \partial A_{ij}^k$. Multiplying these weights by the corresponding activation maps and summing over channels, as in $\mathrm{CAM}_c$, highlights the spatial regions that the model relies on most for predicting class $c$.

We use this mechanism to compare the behavior of the discovered circuits with that of the full model. Following common practice in vision models, we apply Grad-CAM to the last convolutional layer of the GTSRB networks. For visualization, we compute, normalize, and upsample the resulting Grad-CAM maps. Figure 9 illustrates these maps for an illustrative GTSRB sample depicting a roundabout sign.

While the sampling-based circuit is larger than the provably robust one (35 convolutional channels compared to 26), the latter exhibits a closer match to the full model in the final convolutional layer. As shown in Fig. 9b, the heatmaps of the full model on this sample align well with those of the provably robust circuit, whereas the sampling-based circuit shows a less aligned activation pattern, with some loss of emphasis on regions in the sign interior.

In addition, despite the very small perturbation radius (which makes the clean and perturbed images visually almost indistinguishable; Fig. 9a), Fig. 9b shows that the sampling-based circuit shifts its attention between the two inputs, while the provably robust circuit exhibits essentially no variation. This may suggest that the provably robust circuit better maintains its focus under perturbations.

These observations, though not central to our evaluation, provide an additional qualitative lens on how the discovered circuits operate.

Input
**x**

Perturbed input
$\mathbf{z} \in B_{\epsilon_p}^p(\mathbf{x})$

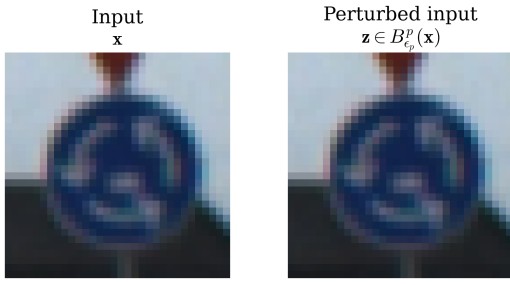

(a) Original input, and a perturbed input in the GTSRB
dataset.

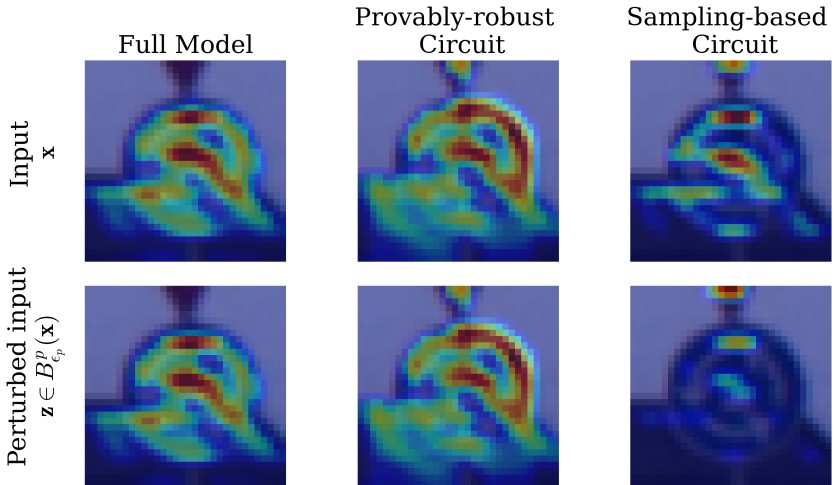

(b) GradCAM computations at the last convolutional layer

Figure 9: Grad-CAM heatmaps comparison for a GTSRB input and its adversarial counterpart, shown at the last convolutional layer across three models (the full model, the provably robust circuit, and the sampling-based circuit).

### E.2.7  ADDITIONAL QUALITATIVE INTERPRETATION OF COMPONENT-LEVEL BEHAVIOR

Another possible direction for qualitative analysis is to assign semantic interpretations to inner components and subgraphs. This may include examining their behaviour and inferring the causal pathways in which they participate.

In the example shown in Figure 4, we consider a CIFAR-10 bird sample together with its adversarial perturbation (also displayed in Figure 10), and compare the two circuit variants extracted from the ResNet model: the sampling-based circuit and the provably robust one. As illustrated, several filter-level components are preserved in the provably robust circuit, enabling it to satisfy the robustness criterion under perturbations.

To analyze the additional components, we focus on the first convolutional layer, as shown in Figure 4. While later-layer interactions could also be insightful, for simplicity and clarity, we restrict our attention to the first-layer filters applied to the perturbed bird image. As the figure illustrates, this layer contains three filters shared by both circuits and one additional filter present only in the provably-robust circuit.

We next examine the clean and adversarial images and their corresponding normalized difference heatmap, presented in Figure 10 within the main paper. Although the perturbation at $\epsilon = 0.015$ is visually almost indistinguishable from the clean input, visualizing their difference reveals that substantial portions of the perturbation concentrate in the lower part of the image, beneath the bird's contour.

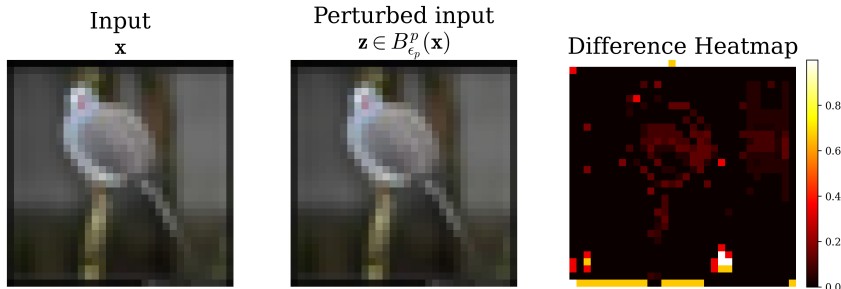

Figure 10: Clean input $\mathbf{x}$, its perturbed version $\mathbf{z} \in B^p_{\epsilon_p}(\mathbf{x})$ with $\epsilon_p = 0.015$, and the corresponding difference heatmap.

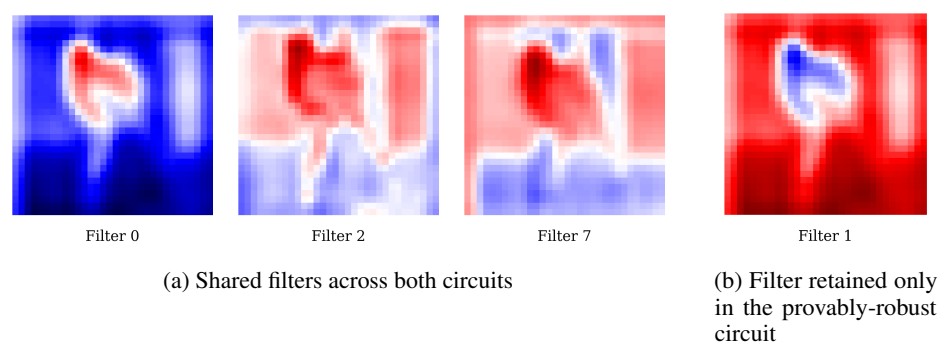

(a) Shared filters across both circuits      (b) Filter retained only in the provably-robust circuit

Figure 11: (a) Activation maps of the shared first-layer filters present in both the provably robust and sampling-based circuits. (b) Activation map of the additional first-layer filter included only in the provably robust circuit. Blue indicates negative activations; red indicates positive activations. White denotes neutral or near-zero values.

To gain further insight, we inspect the activation maps of these filters on the perturbed input (Figure 11). We examine the three filters shared by both circuits as well as the additional filter unique to the provably robust circuit. For each filter, we extract the signed activations (since this layer does not apply a ReLU), normalize them, and upsample them for visibility. The resulting sign-normalized maps display negative activations in blue and positive activations in red.

Across the three shared filters, we observe in Figure 11a that, although they react to and capture aspects of the bird's contour, all of them assign negative values to the lower region where the noise is concentrated. Filter 0 outputs strongly negative values in this area, while filters 2 and 7 produce values between negative and neutral, passing only a weak signal over that region.

In contrast, the additional filter included appears only in the provably-robust circuit (Figure 11b) produces strong positive activations precisely over the lower, noise-affected region. It is the only filter in this layer to do so. This suggests that its inclusion, together with the other filters, may help enrich and stabilize the signal over the perturbed region of the input, potentially contributing to the circuit's certified robustness under this perturbation. Such an illustrative view suggests a possible connection between the retained components and the circuit's robustness.

## F    PATCHING ROBUSTNESS CERTIFICATION

In this experiment, we evaluate the robustness of discovered circuits when non-circuit components are patched with feasible activations drawn from a *continuous* input range, rather than fixed constants, as defined in Section 3.2. Operationally, for any circuit $C$, we test whether perturbing non-circuit components within the range of activations induced by inputs $\mathbf{z} \in \mathcal{B}^\infty_{\epsilon_p}(\mathbf{x})$ can cause a violation of some metric $\|\cdot\|_p$ with tolerance $\delta$. If no such violation exists, we declare $C$ to be *patching-robust* 2.

We compare three patching schemes within the iterative discovery framework (Algorithm 1):

1. **Zero patching:** sets all non-circuit components to zero.

2. **Mean patching:** replaces non-circuit components with their empirical means, estimated from 100 randomly selected training samples.

3. **Provably robust patching:** verifies robustness across the full range of feasible activations induced by a continuous input domain (Def. 2).

## F.1 METHODOLOGY

### F.1.1 PATCHING SIAMESE NETWORK FOR VERIFICATION

As an interface for verifying patching robustness, we employ a *Patching Siamese* network with two branches: (i) a full-network *patching branch*, used to capture non-circuit activations for patching; and (ii) a *circuit branch* restricted to $C$, where every non-circuit component is replaced by the activation of its counterpart in the patching branch. This replacement is implemented through dedicated wiring that copies the required activations from the first branch to the second.

We examine activations induced by inputs from a neighborhood around $\mathbf{x}$: $\mathcal{Z} = \mathcal{B}_{\epsilon_p}^\infty(\mathbf{x})$. As outlined in Section 3.2, the siamese network is fed a concatenated input $(\mathbf{x}, \mathbf{z})$ (along the feature axis for MNIST and the channel axis for CNNs). Here, $\mathbf{x}$ is routed to the circuit branch and $\mathbf{z}$ to the patching branch. The verification domain is applied only to $\mathbf{z}$, while $\mathbf{x}$ is held fixed. This setup simulates the circuit running on $\mathbf{x}$, with its non-circuit activations replaced by those induced from $\mathbf{z} \in \mathcal{Z}$. We note that, for numerical stability on the verifier's side, the circuit-branch input is enclosed in a negligible $10^{-5}$ $L^\infty$ ball.

For the output criterion, the resulting circuit logits are verified against the logits of the full model, $f_G(\mathbf{x})$ (precomputed independently of the Siamese construction) under the logit-difference metric with tolerance $\delta$. This guarantees that the patching-robustness property (Def. 2) holds.

**Input Neighborhoods.** As in Section E.1.2, we define the neighborhood $\mathcal{Z} = \mathcal{B}_{\epsilon_p}^\infty(\mathbf{x})$ using the $\ell_\infty$ norm.

## F.2 ROBUSTNESS EVALUATION AND PARAMETER VARIATIONS

After discovery (using zero, mean, or provably robust patching), we verify the resulting circuits with the Patching Siamese Network over the same $\mathcal{B}_{\epsilon_p}^\infty(\mathbf{x})$, reporting circuit size, runtime, and patching robustness. Since typical $\epsilon_p$ values in the literature target *input* perturbations, we use larger values for the patching domain to reflect the broader variability of internal activations while avoiding off-distribution regimes. We report results below for varying $(\epsilon_p, \delta)$ to assess their effects on robustness and size.

### F.2.1 VERIFICATION AND EXPERIMENTAL SETUP (PATCHING ROBUSTNESS)

We use the same hardware configuration as in the input-robustness study E.1.2. We set a Branch-and-bound timeout of **45** seconds for MNIST, GTSRB, and TaxiNet as in the input experiment. For CIFAR-10, iterative discovery queries in the provably robust method are limited to **45** seconds, while discovered circuit-robustness evaluations are allowed up to **120** seconds. Queries that do not complete within these limits are reported as *unknown*.

As in the input robustness experiment E, for fairness, we exclude cases where the robustness check for zero or mean patching timed out from the reported robustness statistics. In our main results, the timeout rates were 12% for MNIST, 2% for TaxiNet, 6.2% for GTSRB, and 31% for CIFAR-10, while in the variations over $\epsilon_p$ ( F.2.2) they averaged 0.5% for TaxiNet and 8.8% for MNIST. Over the $\delta$ variation ( F.2.3), the average timeout rate was 1.5% on TaxiNet and 14% on MNIST.

### F.2.2 VARIATION OF PATCHING NEIGHBORHOOD SIZE $\epsilon_p$

We fix the tolerance level $\delta$ and vary the patching neighborhood size $\epsilon_p$. For CIFAR-10 we use $\delta=0.1$, for MNIST $\delta=0.5$, for TaxiNet $\delta=0.92$, and for GTSRB $\delta=2.0$. Table 12 extends the main results with additional $\epsilon_p$ variations on the MNIST and TaxiNet benchmarks.

| Dataset | $\epsilon_p$ | Zero Patching | | | Mean Patching | | | Provably Patching-Robust Patching | | |
|---|---|---|---|---|---|---|---|---|---|---|
| | | Time (s) | Size ($|C|$) | Rob. (%) | Time (s) | Size ($|C|$) | Rob. (%) | Time (s) | Size ($|C|$) | Rob. (%) |
| MNIST ($\delta=0.5$) | | | | | | | | | | |
| | 0.005 | 0.054 ± 0.182 | 19.87 ± 1.55 | **87.0** ±3.4 | 0.013 ± 0.001 | 19.19 ± 1.87 | **93.0** ±2.6 | 671.08 ±36.86 | 11.32 ± 2.56 | **100.0** ±0.0 |
| | 0.009 | 0.013 ±0.000 | 19.92 ±1.51 | **65.9** ±5.0 | 0.013 ±0.001 | 19.13 ± 1.88 | **64.8** ±5.0 | 583.59 ±44.99 | 16.75 ± 2.26 | **100.0** ±0.0 |
| | **0.010** | 0.060 ±0.322 | 19.96 ±1.50 | **58.0** ±5.3 | 0.016 ±0.003 | 19.16 ±1.84 | **55.7** ±5.3 | 714.87 ±207.08 | 17.03 ±2.30 | **100.0** ±0.0 |
| | 0.050 | 0.015 ±0.005 | 19.76 ±1.50 | **1.2** ±1.2 | 0.015 ±0.003 | 19.09 ±1.81 | **1.2** ±1.2 | 598.41 ±156.96 | 23.20 ±1.17 | **100.0** ±0.0 |
| TaxiNet ($\delta=0.92$) | | | | | | | | | | |
| | 0.005 | 0.061 ±0.183 | 5.78 ±0.79 | **93.0** ±2.6 | 0.022 ±0.061 | 5.38 ±0.65 | **100.0** ±0.0 | 220.67 ±57.23 | 4.60 ±0.74 | **100.0** ±0.0 |
| | 0.008 | 0.026 ±0.097 | 5.78 ±0.79 | **62.0** ±4.9 | 0.008 ±0.001 | 5.38 ±0.65 | **77.0** ±4.2 | 168.99 ±44.83 | 5.26 ±0.54 | **100.0** ±0.0 |
| | **0.010** | 0.024 ±0.059 | 5.78 ±0.78 | **57.1** ±5.0 | 0.025 ±0.068 | 5.39 ±0.65 | **63.3** ±4.9 | 175.73 ±52.71 | 5.41 ±0.59 | **100.0** ±0.0 |
| | 0.030 | 0.009 ±0.002 | 5.78 ±0.79 | **27.0** ±4.4 | 0.008 ±0.001 | 5.38 ±0.65 | **40.0** ±4.9 | 95.31 ±15.54 | 6.04 ±0.20 | **100.0** ±0.0 |
| | 0.050 | 0.024 ±0.058 | 5.77 ±0.78 | **0.0** ±0.0 | 0.012 ±0.032 | 5.37 ±0.65 | **0.0** ±0.0 | 89.37 ±17.58 | 7.07 ±0.26 | **100.0** ±0.0 |
| CIFAR-10 ($\delta=0.1$) | | | | | | | | | | |
| | **0.030** | 0.109 ± 0.321 | 65.07 ± 3.00 | **46.4** ± 6.0 | 0.046 ± 0.003 | 64.07 ± 3.60 | **33.3** ± 5.7 | 5408.51 ± 1091.05 | 65.55 ± 1.64 | **100.0** ± 0.0 |
| GTSRB ($\delta=2.0$) | | | | | | | | | | |
| | **0.005** | 0.284 ± 0.951 | 32.65 ± 4.24 | **38.0** ± 4.4 | 0.041 ± 0.009 | 33.40 ± 4.16 | **40.5** ± 4.5 | 2907.17 ± 721.67 | 34.34 ± 4.07 | **100.0** ± 0.0 |

Table 12: Variations on patching neighborhood size $\epsilon_p$ with fixed tolerance $\delta$. Reported values are means with standard deviations (formatted as {mean ± std}). For robustness (a binary outcome), we report the mean robustness with its standard error (SE), and display robustness values in **bold**. Rows highlighted in gray correspond to the results selected in the main paper.

### F.2.3 VARIATION OF TOLERANCE TOLERANCE $\delta$

We fix $\epsilon_p$ and examine various tolerance values. As in the main paper results, we use $\epsilon_p=0.01$ for MNIST and Taxinet and vary the tolerance $\delta$. Results are reported in Table 13

| Dataset | $\delta$ | Zero Patching | | | Mean Patching | | | Provably Robust Patching | | |
|---|---|---|---|---|---|---|---|---|---|---|
| | | Time (s) | Size ($|C|$) | Rob. (%) | Time (s) | Size ($|C|$) | Rob. (%) | Time (s) | Size ($|C|$) | Rob. (%) |
| MNIST ($\epsilon_p=0.01$) | | | | | | | | | | |
| | 0.25 | 0.142 ± 0.476 | 21.36 ± 1.87 | **48.6** ± 5.9 | 0.013 ± 0.000 | 21.22 ± 1.74 | **52.8** ± 5.9 | 523.67 ± 41.31 | 19.42 ± 2.01 | **100.0** ± 0.0 |
| | **0.50** | 0.060 ± 0.322 | 19.96 ± 1.50 | **58.0** ± 5.3 | 0.016 ± 0.003 | 19.16 ± 1.84 | **55.7** ± 5.3 | 714.87 ± 207.08 | 17.03 ± 2.30 | **100.0** ± 0.0 |
| | 1.00 | 0.013 ± 0.000 | 16.88 ± 1.85 | **61.2** ± 4.9 | 0.013 ± 0.001 | 16.11 ± 1.93 | **66.3** ± 4.8 | 690.42 ± 44.38 | 11.37 ± 2.64 | **100.0** ± 0.0 |
| TaxiNet ($\epsilon_p=0.01$) | | | | | | | | | | |
| | 0.50 | 0.011 ± 0.003 | 6.86 ± 0.78 | **74.0** ± 4.4 | 0.010 ± 0.001 | 5.83 ± 0.40 | **80.0** ± 4.0 | 157.67 ± 33.68 | 6.00 ± 0.00 | **100.0** ± 0.0 |
| | 0.80 | 0.080 ± 0.210 | 6.04 ± 0.77 | **57.1** ± 5.0 | 0.010 ± 0.001 | 5.49 ± 0.54 | **61.2** ± 4.9 | 204.38 ± 40.04 | 5.68 ± 0.55 | **100.0** ± 0.0 |
| | **0.92** | 0.024 ±0.059 | 5.78 ±0.78 | **57.1** ±5.0 | 0.025 ±0.068 | 5.39 ±0.65 | **63.3** ±4.9 | 175.73 ±52.71 | 5.41 ±0.59 | **100.0** ±0.0 |
| | 1.20 | 0.011 ± 0.001 | 5.44 ± 0.96 | **46.9** ± 5.0 | 0.010 ± 0.003 | 5.20 ± 0.79 | **58.2** ± 5.0 | 246.14 ± 50.82 | 5.15 ± 0.51 | **100.0** ± 0.0 |

Table 13: Variation of tolerance level $\delta$ with fixed patching neighborhood size $\epsilon_p = 0.01$. Reported values are means with standard deviations (formatted as mean ± std). For robustness (a binary outcome), we report the mean robustness with its standard error (SE), and display robustness means in **bold**. Rows highlighted in gray correspond to the results selected in the main paper.

## G EXPLORING CIRCUIT MINIMALITY GUARANTEES

In this experiment, we examine circuits that must simultaneously satisfy both input-robustness (Def. 1) and patching-robustness (Def. 2), as introduced in Section 3.2. Specifically, we define $\Phi$ to require that circuits remain robust within the input neighborhood $\mathcal{Z} = \mathcal{B}_{\epsilon_{\text{in}}}^{\infty}(\mathbf{x})$, when non-circuit components are patched with values drawn from the patching neighborhood $\mathcal{Z}' = \mathcal{B}_{\epsilon_{\text{p}}}^{\infty}(\mathbf{x})$.

Thus, two domains are involved: one for inputs and one for obtaining activations used in patching. Here, $\epsilon_{\text{in}}$ and $\epsilon_{\text{p}}$ denote the radii of the respective $L^{\infty}$-balls. In our setup, we use $\epsilon_{\text{in}} = 0.01$, $\epsilon_{\text{p}} = 0.012$, and $\delta = 2.0$.

### G.1 VERIFYING SIMULTANEOUS INPUT- AND PATCHING-ROBUSTNESS WITH TRIPLED SIAMESE

We certify simultaneous input- and patching-robustness using a *tripled Siamese network* with three branches, each evaluated on its designated domain: (i) a full-network *patching branch*, which

processes inputs $\mathbf{z}' \in \mathcal{B}_{\epsilon_p}^\infty(\mathbf{x})$ to capture activations for use as patching values; (ii) the *full network*, which processes inputs $\mathbf{z} \in \mathcal{B}_{\epsilon_{in}}^\infty(\mathbf{x})$ to provide reference logits $f_G(\mathbf{z})$; and (iii) the *circuit branch*, also evaluated on $\mathbf{z} \in \mathcal{B}_{\epsilon_{in}}^\infty(\mathbf{x})$ but restricted to $C$, where non-circuit components are masked and instead receive transplanted activations from the patching branch.

This special wiring enables the direct transfer of non-circuit activations, allowing the verifier to certify that the circuit logits $f_C(\mathbf{z})$ remain faithful to $f_G(\mathbf{z})$ across the input neighborhood $\mathcal{B}_{\epsilon_{in}}^\infty(\mathbf{x})$ under patching values induced by $\mathcal{B}_{\epsilon_p}^\infty(\mathbf{x})$, thereby establishing the simultaneous input- and patching-robustness property.

We evaluate three discovery strategies under this setting, all applied with the combined robustness predicate $\Phi$ defined above:

1. **Iterative discovery:** Algorithm 1.
2. **Quasi-minimal search:** Algorithm 4.
3. **Blocking-sets MHS method:** Algorithm 2, leveraging circuit blocking set duality to approximate cardinally minimal circuits.

### G.2 BLOCKING-SETS MHS METHOD: ANALYSIS AND EXPERIMENTAL DETAILS

**Experimental setup.** We conduct experiments on the MNIST network (34 hidden neurons). Since contrastive subsets are enumerated in increasing order of size, the number of verification queries grows combinatorially. Even when restricted to subset sizes $t \in \{1, 2, 3\}$, each batch requires thousands of verification calls. To keep computations tractable, we evaluate singletons ($k = 1$ per batch) and enforce a 30-second timeout per query. In one rare case, the procedure produced an *empty circuit* (size 0), as $\Phi$ held vacuously under a particular choice of environments and metric, eliminating all components. This case was excluded from the reported results.

**Parallelism.** Unlike iterative discovery, where elimination steps are sequentially dependent, the verification of contrastive subsets is independent. This independence allows full parallelization: we distribute verification queries across 14 workers, with runtime scaling nearly linearly with the number of workers.

**Properties.** Under monotonic $\Phi$, the MHS method yields either (i) a lower bound on the size of any cardinally minimal circuit, or (ii) when the hitting set itself satisfies $\Phi$, a certified cardinally minimal circuit. Although more computationally expensive than iterative discovery, MHS provides strictly stronger guarantees: if the hitting set is valid, the result is provably cardinally minimal; otherwise, its size gives a tight lower bound that exposes whether iterative discovery reached cardinal minimality and quantifies any gap.

## H DISCLOSURE: USAGE OF LLMS

An LLM was used solely as a writing assistant to correct grammar, fix typos, and enhance clarity. It played no role in generating research ideas, designing the study, analyzing data, or interpreting results; all of these tasks were carried out exclusively by the authors.

