# OpenReview forum: "Formal Mechanistic Interpretability: Automated Circuit Discovery with Provable Guarantees"
_ICLR.cc/2026/Conference — ICLR 2026 Poster_

### Official Review · Reviewer_WbQb · 2025-10-26

**Soundness:** 3
**Presentation:** 3
**Contribution:** 3
**Rating:** 6
**Confidence:** 4

**Summary:**

This paper proposes a framework for provable circuit discovery in mechanistic interpretability, offering formal guarantees of input robustness, patching robustness, and minimality. It introduces two algorithms—a greedy local-minimality search and a blocking-set duality method—for discovering circuits with verified faithfulness using α–β-CROWN. Experiments on MNIST, CIFAR-10, GTSRB, and TaxiNet confirm 100 % robustness within specified domains but show high computational cost.

**Strengths:**

1. Clear definitions and proofs for robustness and minimality in circuit discovery.

2. The greedy and hitting-set approaches are well motivated and practically implementable.

3. Connects mechanistic interpretability with formal verification through provable guarantees.

**Weaknesses:**

1. High computational cost: Verification with α–β-CROWN for each candidate circuit is extremely slow; scalability remains a bottleneck.

2. Small experimental scope: Only small convolutional and MLP networks are tested; no evidence on larger or modern architectures.

3. Limited interpretive analysis: The paper emphasizes correctness and robustness, but offers little discussion of what the discovered circuits mean semantically.

4. Evaluation imbalance: Most comparisons are against heuristic baselines without runtime or coverage trade-offs clearly analyzed.

**Questions:**

A recent paper, “Learning Minimal Neural Specifications” (Geng et al., NeuS 2025), finds that a small subset of neurons can often characterize a model’s robust behavior. Your minimal-circuit formulation appears conceptually related. Could you discuss the connections between these phenomena?

---

> ### Author Response · Authors · 2025-11-21
>
> We thank the reviewer for their thorough and constructive feedback and for acknowledging the significance of our work.
>
> **Additional coverage and runtime analysis**
>
> In response to these valuable suggestions by the reviewer, we have included an additional experiment in Appendix E.2.4 that conducts a coverage analysis of the sampling-based baseline versus the provably robust method. We examine several increasing input-robustness radii and report the structural relationship between the two circuits in terms of overlap and differences. We use standard symmetric coverage metrics (IoU, Dice) as well as asymmetric ones, reporting their means and standard deviations, and complement this with a quantitative breakdown of differences in Table 3. Overall, the results indicate that the provably robust circuits recover most of the sampling-based units at small perturbations, but the overlap steadily decreases as the perturbation size grows. Given the time constraints of the rebuttal period, we have conducted this analysis only on MNIST for now and will incorporate the remaining benchmarks in the final version.
>
> To further address rthe untime behavior of the provably robust method, we have included an additional analysis over a range of continuous input neighborhoods with increasing perturbation radii. For each perturbation level (input environment), we track the circuit size as a function of elapsed time and report the corresponding curves, summarizing them with mean and standard deviation in a dedicated section (Appendix E.2.5). Figure 8 illustrates the pace of the provably-robust circuit discovery across different input environments, showing that larger neighborhoods are associated with a slower reduction in circuit size over time and consistently increased runtimes. Here as well, we will include the remaining benchmark evaluations in the final version.
>
> **Computational barriers of certification and future directions**
>
> We agree with the reviewer that our method, like any approach that provides certifiable guarantees over continuous domains, relies on neural network verification. Verification techniques have made remarkable progress in scalability in recent years (see, e.g., 1-6), but they are indeed still not suited for state-of-the-art models yet. However, and as several reviewers have also emphasized, these computational barriers arise from the certification tools themselves rather than from our circuit-discovery framework. Continued advances in verification will naturally broaden the class of models to which our method applies.
>
> Beyond this, we believe that the theoretical and algorithmic contributions of our work, highlighted by the reviewer as well as the other reviewers, offer a more rigorous foundation for discussing the circuit discovery problem within the MI community. Our analysis of input-, patching-, and minimality-based guarantees, along with the connections we prove that hold among them, helps formalize some central aspects of circuit discovery. We therefore hope that this work will influence future attempts to scale such guarantees, whether by certifying sub-components of larger models, which may be more tractable yet still offer meaningful guarantees, or by developing probabilistic or statistical analogues of our guarantees that may be weaker but more tractable. In this sense, we view our work as a strong baseline for exploring these directions.
>
> **Connections to the work of Geng et al. (NeuS 2025)**
>
> We thank the reviewer for pointing us to this recent work, which we have now included in our related-work section. Geng et al. study activation-pattern specifications (i.e., sets of active/inactive neuron states) as a means to identify more meaningful input regions for robustness verification. Their method aims to find a minimal activation pattern (a set of active/inactive neuron constraints) that, when used as constraints during verification, will enable certifying the *full original model* over a more meaningful input region.
>
> While their approach is also rooted in verification and seeks minimality, its goal differs from ours: we aim to extract *circuits* - actual subgraphs of the model that run as standalone models by themselves and capture causal computational pathways. This objective introduces its own set of challenges, including comparing the behavior of the full model with that of the extracted circuit, handling multiple forms of *patching*, and reasoning about minimality in a way that accounts for the different input- and patching-robustness guarantees we analyze in our four minimality notions.
>
> We believe there can be room for some interesting potential future work bridging these two directions, for example, attempting to apply some of our minimality and algorithmic notions to activation-patterns, or using constrained activation patterns to certify additional regions of the latent space in our extracted circuits.

---

> ### Author Response · Authors · 2025-11-21
>
> **Qualitative perspectives of the obtained provably robust circuits**
>
> Since our work focuses on providing formally provable guarantees for circuit discovery, our contributions were designed to tackle the theoretical, systematic, and algorithmic aspects of the problem. Accordingly, our empirical evaluation has centered on examining circuits under input and patching perturbations, demonstrating how sampling-based methods can violate circuit-faithfulness criteria, and showing that robustness certification can help validate these properties. That said, we fully agree that exploring how these circuit-level guarantees relate to the functional or semantic behavior that circuits capture is an especially exciting direction for future research. We believe our results lay a strong basis for this line of work.
>
> To help motivate this direction, we added two small qualitative examples to the appendix. The first, following a suggestion by reviewer ZmUt, visualizes the activation maps of all four filters in the first convolutional layer of the circuits shown in Figure 1. As presented in Appendix E.2.7, filters f0, f2, and f7 (that appear in both circuits) exhibit their strongest positive responses on the bird object itself, while remaining neutral or negative in the background. The additional filter f1, which is present only in the provably robust circuit, in contrast, shows strong positive activations in the background region, especially along the lower part of the image, where the adversarial perturbation is concentrated. This indicates that f1 is specifically sensitive to the background structure around the bird, responding strongly to the contrast between the object and its surroundings, and may function as a stabilizing component that encodes additional information about this external region. Notably, its activation is strongest especially in the area where the perturbation violates the robustness guarantee, suggesting that f1 helps shield the circuit against this vulnerability.
>
> In a second qualitative example, we visualize a phenomenon we observed across multiple derived circuits (Appendix E.2.6). Specifically, we compare GradCAM [7] attributions for a circuit obtained via our provably robust method versus one obtained via the sampling-based approach. Although both circuits are of comparable size (in this case, the provably robust circuit is even smaller), the robust circuit’s attribution map is visibly far closer to that of the original model. This suggests that, at least under the GradCAM criterion, the provably robust circuit more faithfully reflects the model’s internal representation of images of this type than the sampling-based circuit. Even more interestingly, under an infinitesimally small perturbation of $\epsilon=0.001$, the sampling-based circuit’s GradCAM attribution changes dramatically, while the provably robust circuit remains stable. This highlights that the sampling-based method failed to generalize even under small input perturbations, whereas the provably robust approach naturally avoids this issue. This observation is also consistent with the broader set of quantitative results reported in our experimental section.
>
> [1] First Three Years of the International Verification of Neural Networks Competition (VNN-COMP) (Brix et al., STTT 2023)
>
> [2] The Fifth International Verification of Neural Networks Competition (vnn-comp 2024): Summary and results (Brix et al., STTT 2024)
>
> [3] Beta-Crown: Efficient Bound Propagation with Per-Neuron Split Constraints for Neural Network Robustness Verification (Wang et al., NeurIPS 2021)
>
> [4] Scalable Neural Network Verification with Branch-and-bound Inferred Cutting Planes (Zhou et al., NeurIPS 2024)
>
> [5] SDP-CROWN: Efficient Bound Propagation for Neural Network Verification with Tightness of Semidefinite Programming (Chiu et al., ICML 2025)
>
> [6] Clip-and-Verify: Linear Constraint-Driven Domain Clipping for Accelerating Neural Network Verification (Zhou et al., NeurIPS 2025)
>
> [7] Grad-CAM: Visual Explanations From Deep Networks via Gradient-Based Localization (Selvaraju et al., CVPR 2017)

---

> ### Comment · Reviewer_WbQb · 2025-11-22
>
> I appreciate the authors’ detailed rebuttals and clarifications. The work is interesting and is supported by both theoretical analysis and empirical evidence. I will therefore maintain my positive evaluation and score.

---

### Official Review · Reviewer_X5ZF · 2025-10-29

**Soundness:** 4
**Presentation:** 4
**Contribution:** 4
**Rating:** 8
**Confidence:** 2

**Summary:**

In the context of mechanistic interpretability for neural networks, the authors use neural network verification methods to come up with circuits with provable guarantees. Authors claim to outperform conventional circuit discovery methods., i.e.e they provide stronger robustness guarantees for discovered circuits.

**Strengths:**

- Problem addressed is impactful and the authors identified clear blind spot in literature
- Thorough theoretical contribution
- Experiments rely on VNN-COMP community-benchmark. Results show authors method strongly outperforms the chosen baselines, across all experiments.
- Paper is clear and flows well. Authors contribution also clear.
- Literature review carried out with diligence. I am not a domain expert - fellow reviewers with greater expertise may have identified gaps.

**Weaknesses:**

- Some neural network verification concepts could have been introduced more in details (e.g. "Patching")
- Lack of running example make reading hard to instantiate into a real-world, impactful use case.

**Questions:**

- I could not find a discussion on the computational overhead of the method w.r.t. the baselines.  Could you briefly post a comment about it? (Apologies if I have missed)

---

> ### Author Response · Authors · 2025-11-21
>
> We thank the reviewer for their thorough and constructive feedback and for acknowledging the significance of our work.
>
>
> **Incorporating running examples within the paper**
>
> We agree with the reviewer that integrating running examples can significantly strengthen the presentation. Following this suggestion, we have added three new examples and visualizations: one illustrating the Siamese encodings used for certifying input robustness (Fig. 1), and another depicting the encoding of the patching-robustness query (Fig. 2); both now appear in Section 3. In addition, we brought a detailed running example, previously only included in the appendix, into the main text to clarify the four *minimality* guarantees formalized in our work. We believe these additions improve the overall flow and clarity of the paper, and we thank the reviewer for the helpful suggestion. Moreover, in the final version, we also plan to include a motivating real-world use case.
>
> Lastly, following suggestions from reviewers ZmUt and WbQb, we added two small qualitative examples in Appendix E.2.6 and E.2.7. The first example highlights a potential role of filter f1 in Figure 1 - present only in the provably robust circuit but absent from the sampling-based one - as a contour detector for the object’s outer shape. Its strongest positive responses appear in the lower region of the image, which coincides with the area where the perturbation vulnerability arises. The second compares GradCAM [1] attributions of the provably robust circuit with those of the sampling-based circuit, showing that the provably robust circuit more closely matches the original model’s attributions and remains stable under small input perturbations, unlike the sampling-based circuit. We hope these examples also help motivate future real-world investigations of circuits with provable guarantees.
>
>
>
>
>
>
> **Adding background on neural network verification and patching**
>
>
> Thank you for this suggestion. Following this remark, we have added, in addition to the main paper background on patching (Sec. 2.1) and neural network verification (Sec. 2.2), a more in-depth background on these concepts in Appendix A as well. We believe these additions will provide readers with important and relevant context.
>
> **A discussion on the computational overhead of the method**
>
> As discussed in Section 4, the iterative procedure (Algorithm 1) performs $\mathcal{O}(|G|)$ evaluations of the predicate $\Phi(C,G)$, which serves as a general test for whether a circuit $C$ is faithful with respect to $G$. In sampling-based MI approaches, $\Phi$ typically corresponds to drawing datapoints and running them through the model. In contrast, in the provably-robust certification setting, each invocation of $\Phi$ requires issuing a neural-network certification query, which is more computationally expensive, but essential for obtaining formally certified robustness instead of relying on sampling.
>
> It is also important to note that the $\mathcal{O}(|G|)$ versus $\mathcal{O}(\log(|G|))$ number of $\Phi$-queries reflects a design choice regarding the desired degree of minimality. The *quasi*-minimal variant is more efficient and requires only a logarithmic number of queries, whereas the *locally*-minimal variant achieves a stronger minimality guarantee at the cost of a linear number of evaluations. The even stronger notions, of subset- or cardinal-minimality, arise from the monotonicity of the predicates, which enables the algorithm to converge to increasingly stronger minima. Algorithm 2 targets the strongest form, cardinal minimality, but this is also the most computationally demanding, as it may require computing many blocking sets in the worst case. Nevertheless, as our empirical results show, restricting this parameter to a small value already performs very well in practice. Section 5.3 illustrates several of these trade-offs. Following the reviewer’s suggestion, we will improve the discussion of these points and highlight them more prominently in the main text of the final version.
>
> [1] Grad-CAM: Visual Explanations From Deep Networks via Gradient-Based Localization (Selvaraju et al., CVPR 2017)

---

### Official Review · Reviewer_ZmUt · 2025-10-31

**Soundness:** 4
**Presentation:** 4
**Contribution:** 4
**Rating:** 8
**Confidence:** 4

**Summary:**

This paper focuses on automated circuit discovery in neural networks, one of the key challenges of mechanistic interpretability (MI). Current circuit discovery methods are heuristic and rely on sampling and approximations, therefore failing to provide theoretical guarantees about the obtained circuits (faithfulness and robustness to perturbations). The authors introduce a novel framework that uses neural network verification testing to discover circuits with three types of provable guarantees:
- Input-domain robustness, ensuring that the circuit's behavior remains faithful across a continuous input region.
- Patching-domain robustness, guaranteeing faithfulness under a continuous range of perturbations (patching) to the activations of non-circuit components.
- Minimality: The authors introduce and formalize four types of circuit minimality, hierarchically ordered (quasi-, local, subset- and cardinal minimality) and provide algorithms to either obtain or approximate those.

Some of the key technical contributions of this work include:
- A siamese network encoding, which allows standard verifiers to certify the aforementioned properties
- Identifying a "circuit monotonicity" property that enables stronger minimality guarantees
- Using a circuit-blocking-set duality based on Minimum Hitting Sets (MHS) to find cardinally-minimal circuits.

The authors validate their approach on several vision models and benchmarks, obtaining circuits with substantially stronger robustness guarantees than sampling-based baselines.

**Strengths:**

- The core contribution (applying formal verification to automated circuit discovery) is novel and significant. It directly addresses a fundamental and critical limitation in the field of MI (moving from heuristic approximations to provable guarantees), which has been acknowledged in many recent works in the literature. This work makes a significant step towards increasing the reliability of MI methods and is much-needed.
- The paper is technically excellent. The formalizations are clear, and the introduced hierarchy of minimality guarantees (Definitions 3 through 6) clarifies a concept often used loosely in the literature. All of the stated theorems are proved in depth in the Appendix, which is rare to find in this type of work.
- The experiments clearly support the paper's claims. The authors use the SOTA α,β-CROWN verifier on standard benchmarks, and their proposed methods achieve 100% certified robustness (Tables 1 and 2), whereas sampling-based methods largely fail. Finally, the experiments in section 5.3 are a good illustration of the trade-off between minimality strength (as defined in the proposed hierarchy) and runtime.
- The paper is very well-written, given the complexity of the topic.

**Weaknesses:**

- The most significant weakness (which the authors do acknowledge) is the scalability of the neural network verification methods. The experiments are conducted on relatively small vision models (e.g. ResNet2b) yet already require computation time in the minutes to hour range. Current MI research largely focuses on large LLMs or other models using the Transformers architecture, and as proposed, the authors' framework would likely be computationally intractable for such models. This is an inherent challenge of the verification field and by no means a flaw in the paper's methodology itself and doesn't detract from the foundational contribution of the work, but it does limit the immediate practical applicability to SOTA models.
- The paper could be strengthened by adding a more detailed qualitative analysis of the discovered circuits. For example, the provably robust circuit depicted in Figure 1 contains an extra component. A natural question is: What functional role does it play? An illustrative example would make the benefits of the approach even more tangible and would also be interesting from a research perspective.
- The paper uses a logit-difference metric for faithfulness, which works well for verification and is a standard choice in MI, but may not always accurately reflect the semantic meaning of a circuit that one may be interested in. It is not entirely clear if a small logit difference is always the most meaningful proxy for a circuit "doing the same thing" as the model (a circuit could potentially maintain a small logit difference but modify its internal representations or attributions in a way which may be significant). However, this is, again, the de facto standard used in most MI research and this is a fairly minor point.

**Questions:**

- Have the authors considered adapting the framework to certify different properties? For instance, instead of bounding the logit difference (which can be problematic, as stated above), could it be adapted to certify that the predicted class remains invariant across the input domain? This seems like a natural guarantee for classification.
- The complexity of MHS depends on enumerating blocking sets up to size t_max. What was the practical size of the blocking sets found in the experiments, and how large did t_max need to be to provide good lower bounds or find the minimal circuit? (My apologies if this is answered in the attached codebase, which I did not consult)
- For the example in Figure 1, do the authors have any functional hypothesis for why the components highlighted in green are essential for robustness (handling specific edge cases)? What kinds of input perturbations would cause the sampling-based circuit to fail, which the provably-robust circuit correctly handles?
- Do the authors have any insight into potential ways to apply this framework to larger models/architectures, such as Transformers? Are there specific verification techniques that offer a better trade-off for this application?

---

> ### Author Response · Authors · 2025-11-21
>
> We thank the reviewer for their thorough and constructive feedback and for acknowledging the significance of our work.
>
> **The choice of the logit-difference metric for evaluating faithfulness**
>
> We thank the reviewer for raising this interesting point. As correctly noted by the reviewer, we chose the logit-difference metric because it is widely used in prior MI work and therefore aligns well with the existing literature. Importantly, though, our method is agnostic to the specific output criterion: the metric applied to both the full model and the circuit can be adapted to match the needs of the task. To demonstrate this flexibility, and following the reviewer’s suggestion, we conducted additional experiments (currently on the input-robustness certification task for MNIST), which we have added to the appendix. Alongside the logit-difference metric, we evaluated two alternative criteria:
>
> 1. *A classification-oriented margin metric*, as explicitly (and justifiably) suggested by the reviewer, which enforces that the predicted top class exceeds the runner-up by at least a given margin.
>
> 2. *An elementwise absolute-maximum metric*, which does not enforce class consistency but instead bounds the maximum per-dimension deviation between the circuit and full-model outputs.
>
> Both metrics are now supported in our codebase, and their corresponding results are presented in ``Appendix E.2.3: Evaluation under Alternative Output Metrics’’. Across all evaluated metrics, the trend remains consistent: our provable method provides provably certified robustness while maintaining circuit sizes comparable to those produced by the sampling-based baseline, which yields significantly lower robustness. In the final version of the paper, we will include the full evaluation of these metrics across the additional benchmarks and patching-guarantee settings.
>
> **Scalability of NN formal verification methods and future directions**
>
> As the reviewer rightly noted, our method, like any approach that provides certifiable guarantees over continuous domains, relies on neural network verification. Verification techniques have made remarkable progress in scalability in recent years (see, e.g., 1-6), but they are still not suited for state-of-the-art models. As the reviewer also emphasized, these limitations stem from the certification tools themselves rather than from our circuit discovery framework. Continued advances in verification will naturally expand the range of models to which our method can be applied.
>
> We are optimistic that continued advances in neural network verification will eventually make certification practical for LLMs. In the meantime, we believe our results already enable two concrete research directions that can be pursued: (1) Verification on targeted sub-components of LLMs, such as critical bottlenecks, abstractions, or sub-circuits, may be more tractable while still providing meaningful guarantees. (2) Developing statistical or probabilistic analogues of our guarantees, which may be weaker but more scalable, is another promising avenue. Exploring such analogues naturally raises several interesting questions: How do input-robustness, patching-robustness, and minimality behave under such probabilistic formulations? Under what assumptions can circuit monotonicity be preserved? How could different distributional assumptions affect the results? We view these as exciting avenues for future work and believe our work provides a strong basis for exploring them.

---

> ### Author Response · Authors · 2025-11-21
>
> **Qualitative perspectives of the obtained provably robust circuits**
>
> Our work mainly focuses on providing formally provable guarantees for circuit discovery, and, as the reviewer correctly noted, our contributions were designed to tackle the theoretical, systematic, and algorithmic aspects of the problem. Accordingly, our empirical evaluation has centered on examining circuits under input and patching perturbations, demonstrating how sampling-based methods can violate circuit-faithfulness criteria, and showing that robustness certification can help validate these properties. That said, we fully agree that exploring how these circuit-level guarantees relate to the functional or semantic behavior that circuits capture is an especially exciting direction for future research. We believe our results lay a strong basis for this line of work.
>
> To help motivate this direction, we added two small qualitative examples to the appendix. The first, following the reviewer’s suggestion, visualizes the activation maps of all four filters in the first convolutional layer of the circuits shown in Figure 1. As presented in Appendix E.2.7, filters f0, f2, and f7 (which appear in both circuits) exhibit their strongest positive responses on the bird object itself, whereas the additional filter f1, present only in the provably-robust circuit, is activated positively by the background surrounding the bird. This indicates that f1 responds to the *contrast* transition between the object and its surroundings, and may act as a kind of *contour detector* that encodes spatial information about the object’s external shape and, through its differing signal, may serve as a form of "stabilizer’’. Notably, its positive activation is strongest along the bottom region of the image, which aligns with the adversarial perturbation that violates robustness in precisely that area, suggesting that f1 may help shield the circuit against this vulnerability. We did not include visualizations of deeper-layer filters, as feature dependencies make their roles harder to interpret visually. Nonetheless, we believe this type of functional analysis is an interesting direction for future work.
>
> In a second qualitative example, we aimed to present a phenomenon that we encountered across multiple derived circuits (Appendix E.2.6). Specifically, we compare GradCAM [7] attributions for a circuit obtained via the provably robust method versus one obtained via the sampling-based approach. Although both circuits are of comparable size (in this case, the provably robust circuit is *even smaller*), the robust circuit’s attribution map is visibly far closer to that of the original model. This indicates that, at least under the GradCAM criterion, the provably robust circuit more accurately reflects the model’s internal representation for images of this type than the sampling-based circuit. Even more interestingly, under an infinitesimally small perturbation of $\epsilon=0.001$, the sampling-based circuit’s GradCAM attribution changes dramatically, while the provably robust circuit remains stable. This illustrates that the sampling-based method fails to generalize even under small input perturbations, whereas the provably robust approach naturally sidesteps this issue. This observation also aligns with the broader quantitative results presented in our experimental section.
>
> **The possible values of $t_{max}$**
>
> Indeed, this parameter choice was originally stated only in the appendix, and we have now included it in the main text as well. We restricted our experiments to a relatively small value of $t_{max} = 3$, and yet we already observe very strong approximations at this setting (see Figures 2.a and 2.b). This provides encouraging evidence for the effectiveness of computing MHSs, as motivated in our theoretical section.
>
> [1] First Three Years of the International Verification of Neural Networks Competition (VNN-COMP) (Brix et al., STTT 2023)
>
> [2] The Fifth International Verification of Neural Networks Competition (vnn-comp 2024): Summary and results (Brix et al., STTT 2024)
>
> [3] Beta-Crown: Efficient Bound Propagation with Per-Neuron Split Constraints for Neural Network Robustness Verification (Wang et al., NeurIPS 2021)
>
> [4] Scalable Neural Network Verification with Branch-and-bound Inferred Cutting Planes (Zhou et al., NeurIPS 2024)
>
> [5] SDP-CROWN: Efficient Bound Propagation for Neural Network Verification with Tightness of Semidefinite Programming (Chiu et al., ICML 2025)
>
> [6] Clip-and-Verify: Linear Constraint-Driven Domain Clipping for Accelerating Neural Network Verification (Zhou et al., NeurIPS 2025)
>
> [7] Grad-CAM: Visual Explanations From Deep Networks via Gradient-Based Localization (Selvaraju et al., CVPR 2017)

---

> > ### Comment · Reviewer_ZmUt · 2025-11-24
> >
> > I thank the authors for their detailed and high-quality rebuttal and appreciate the additional experiments regarding alternative output metrics. It directly addresses my question, and it is encouraging to see that the results remain consistent.
> >
> > The new analysis on Figure 1 and the addition of the GradCAM analysis are also much appreciated. The identification of f1 and discussion of its potential role is the kind of interpretability insight that reinforces the motivation for using provably robust methods. It could be interesting to see if these "stabilizer" circuits share characteristics with features found in e.g. adversarially trained networks, to see if the provable discovery process could bias the circuit selection toward robust features rather than predictive ones.
> >
> > Finally, the authors have fully addressed my questions regarding the practical values of t_max and the limitations regarding scalability, and I agree that the scalability bottleneck lies with the verification tools rather than the proposed framework itself. One thing that could be interesting is to see how the core property of circuit monotonicity would work in a probabilistic/statistical setting which the authors mention. For example, adding a specific component may increase the *variance* of the circuit's output, and it is not obvious how that would translate in terms of circuit monotonicity.
> >
> > In light of the rigorous methodology and the comprehensive response to my concerns, I am happy to maintain my positive score. My follow-up points are reaching quite far outside the scope of the initial paper, and do not necessarily need to be addressed. I strongly recommend acceptance.

---

### Meta-Review · Area_Chair_FLHB · 2026-01-07

**Summary:**

This paper propose using existing neural network verification methods to certify automated circuit discovery with provable robustness guarantees. While this is a straightforward application of neural network verification methods to a new application (automated circuit discovery), this is the first work of doing this and all the reviewers are positive about it and mostly satisfied with the rebuttal response. Thus a recommendation of accept.

**Reviewer Concerns:**

* Reviewer ZmUt has concerns on scalability and lacking results on SOTA model, which is still outstanding. However, this is an intrinsic limitation of white-box NN verification methods.
* Reviewer X5ZF's concern on lacking running examples have been addressed
* Reviewer WbQb's concern on scalability and high computation cost is still outstanding. However, this is an intrinsic limitation of white-box NN verification methods.

**Reviewer Scores:**

* Reviewre ZmUt stated to maintain positive score of 8
* Reviewer X5ZF likely to maintain score of 8
* Reviewer WbQb stated to remain positive score of 6

---

### Decision · Program_Chairs · 2026-01-26

Accept (Poster)